# Global vegetation resilience linked to water availability and variability

Taylor Smith [1] ✉ & Niklas Boers [2,3,4]

Quantifying the resilience of vegetated ecosystems is key to constraining both present-day and future global impacts of anthropogenic climate change. Here we apply both empirical and theoretical resilience metrics to remotely-sensed vegetation data in order to examine the role of water availability and variability in controlling vegetation resilience at the global scale. We find a concise global relationship where vegetation resilience is greater in regions with higher water availability. We also reveal that resilience is lower in regions with more pronounced inter-annual precipitation variability, but find less concise relationships between vegetation resilience and intra-annual precipitation variability. Our results thus imply that the resilience of vegetation responds differently to water deficits at varying time scales. In view of projected increases in precipitation variability, our findings highlight the risk of ecosystem degradation under ongoing climate change.

The resilience of ecosystems, i.e., their capacity to resist and recover from external perturbations—natural or anthropogenic—has received increasing attention in recent years[1–5]. Key to this discussion is whether ecosystems can potentially exhibit multiple stable equilibrium states with abrupt transitions between them in response to gradual changes in climatic and environmental conditions. For some regions, it has even been suggested that alternative stable states may co-exist for the same climatic forcings; for example, it is thought that several tropical regions support both a stable rainforest and a stable savanna state for a considerable range of mean annual precipitations[6–8]. The capacity of ecosystems to recover to their previous state after a shock—such as a fire, drought, or deforestation—is a critical open question, particularly in view of the impacts of anthropogenic climate change[9]. Changes in ecosystem function can drastically alter carbon sequestration capacities[10]; for example, the Amazon rainforest appears to have recently turned from a globally relevant carbon sink to a net source of carbon[11]. The potential for abrupt transitions between alternative stable states—and corresponding risks of further carbon emissions—is not only confined to the tropics[12,13], making identifying controls on ecosystem resilience a global concern. Indeed, recent work[4,5,14,15] has shown that many regions are losing vegetation resilience; however, the drivers of spatial heterogeneity in resilience and resilience trends remain unconstrained. In this work, we therefore aim to improve our understanding of possible climatic drivers of vegetation resilience.

A wide body of previous research[16,17] has proposed that the capacity of a system to recover from external shocks, and hence the system's resilience, is closely tied to both the variance and the lag-one autocorrelation (AC1) of time series encoding the dynamics of the system in question[1,18–22]; higher values of AC1 and variance are associated with lower resilience (see Methods). Under some assumptions, it can indeed be shown analytically that the variance and AC1 are related to the recovery rate and hence the resilience of the system in question; an empirical confirmation of these relationships—and thereby an empirical justification for the use of variance and AC1 as proxies for vegetation resilience—has recently been provided using global-scale satellite data[4]. Based on the results from the latter study, we will in the following focus on three different ways of estimating vegetation resilience. The direct empirical recovery rate obtained from fitting an exponential recovery model to vegetation time series after experiencing abrupt perturbations will be compared to the theoretical recovery rate estimates inferred from both variance and AC1 (see Methods). The empirical recovery rate is important because it gives a directly measurable resilience metric for those locations where perturbations occurred; on the other hand, the variance- and AC1-based

[1]Institute of Geosciences, Universität Potsdam, Potsdam, Germany. [2]Potsdam Institute for Climate Impact Research, Potsdam, Germany. [3]Technical University of Munich, School of Engineering & Design, Earth System Modelling, Munich, Germany. [4]Department of Mathematics and Global Systems Institute, University of Exeter, Exeter, UK. ✉e-mail: tasmith@uni-potsdam.de

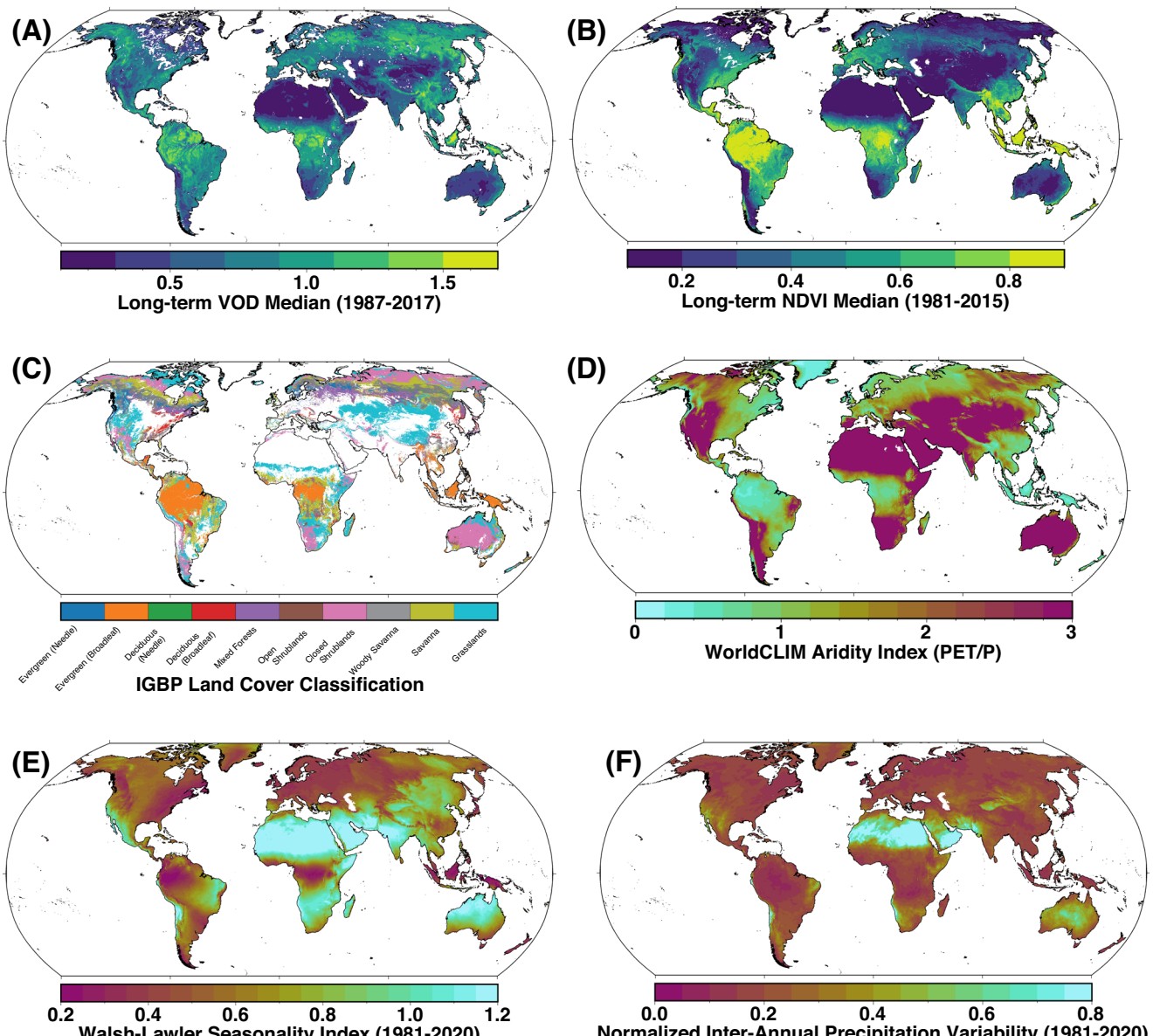

**Fig. 1 | Spatial distribution of data used in this study.** Long-term median **A** vegetation optical depth (VOD, 1987–2017,[23]) and **B** normalized difference vegetation index (GIMMS3g NDVI, 1981–2015,[24]). **C** IGBP Land-cover classes[46], masked for anthropogenic influence (Methods). **D** Global aridity index, adapted from WorldCLIM[31]. Note that higher values correspond to drier conditions. **E** Walsh-Lawler Seasonality Index[32] and **F** normalized inter-annual precipitation variability (Methods) based on ERA5 data (monthly, 1981–2021[33]). See Supplementary Fig. S1 for a similar map of MODIS NDVI.

metrics are important because they yield spatially homogeneous resilience metrics that also cover locations where no empirical recovery rate can be fitted reliably. Moreover, although not done here, variance-and AC1-based estimates in principle allow changes in resilience over time to be quantified.

Verbesselt et al.[1] showed that in the tropics, the AC1 of vegetation systems has an inverse relationship with mean annual precipitation (MAP), suggesting that vegetation in wetter regions is more resilient. Here, we rely on both empirical and theoretical resilience metrics to investigate the effects of water availability and variability—quantified over multiple time scales from seasonal, annual, to multi-annual—on vegetation resilience globally, i.e., for all climate zones and land-cover types.

To investigate global vegetation resilience patterns, we use and compare three different vegetation datasets: long-term satellite-derived vegetation optical depth (VOD) (Fig. 1A)[23], as well as normalized difference vegetation index (NDVI) data from GIMMS3g (Fig. 1B[24])

and MODIS MOD13[25] (see Methods for details on each dataset). Note that the first two vegetation datasets rely upon merging data from different satellite sensors, whereas the latter stems from continuous measurements by a single sensor; potential impacts of the merging of data from different sensors on our results can thereby be controlled. Combining data from different satellite sensors can lead to biases in estimates of the temporal changes of resilience indicators due to induced time-varying changes in the higher-order statistics of the resulting times series[26]. Here, however, we only compute resilience indicators over the full available time spans of each dataset; in this case, a changing satellite composition does not induce systematic biases in our resilience estimates (Methods).

We analyze the dependence of resilience on aridity (Fig. 1D, Methods)—an estimate of water surplus or deficit—and both intra- and inter-annual precipitation variability (Fig. 1E, F, Methods) and also investigate differences in these relationships for varying land-cover types. Importantly, we consider the relationships between the

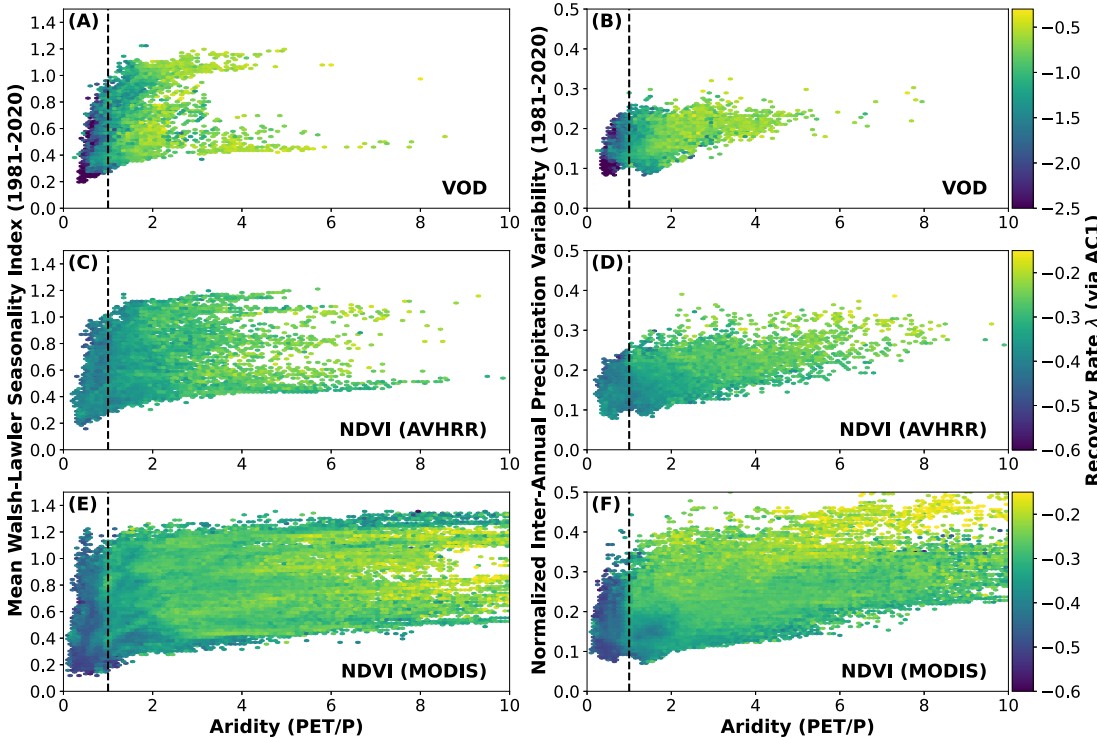

**Fig. 2 | Comparison of aridity and intra- and inter-annual precipitation variability in their relative importance for vegetation resilience at the global scale.** **A**, **B** Vegetation optical depth (VOD), **C**, **D** GIMMS3g normalized difference vegetation index (NDVI), and **E**, **F** MODIS NDVI. Aridity compared to intra-annual (left column) and inter-annual (right column) precipitation variability. Hexbins colored by recovery rate computed from AC1 (minimum five points per bin). Values of the recovery rate $\lambda$ closer to zero imply lower resilience. Transition from water surplus (aridity <1) to deficit marked with dashed vertical line; there is a sharp increase in resilience as water availability increases. Higher inter-annual precipitation variability (right column) consistently leads to lower resilience; intra-annual precipitation variability, i.e., seasonality, has a more varied impact. See Supplementary Fig. S3 for a direct comparison of intra- and inter-annual precipitation variability.

different climatic predictors and vegetation resilience separately for each land-cover type; this assures that differences in resilience indicators caused by different land-cover types are not mistaken for differences in their actual resilience. Our analysis extends the discussion of vegetation resilience and its dependence on long-term precipitation characteristics to the global scale and uncovers succinct and variable relationships between water availability and both theoretical and recently introduced empirical[4] measures of vegetation resilience.

## Results

Vegetation structure and productivity are tightly coupled to both short- and long-term water availability[27]. Differences in annual precipitation sums have been proposed as a control on vegetation resilience; e.g., it has been shown that the AC1 computed from the NDVI in tropical rainforests is negatively correlated with MAP[1], suggesting lower resilience in drier places. However, the relationship between AC1-based resilience estimates and MAP has not been documented globally. Furthermore, vegetation growth and health does not only rely on the amount of water available, but also on the consistency of that water availability[28–30]—even intermittent periods of water deficit will negatively impact plant functioning and growth.

We consider three ways of globally measuring water availability and variability as drivers of vegetation resilience, encompassing multiple overlapping time scales: (1) the Aridity Index (Fig. 1D)[31], which provides a measure of long-term MAP relative to potential evapotranspiration; we consider it more appropriate to consider aridity rather than, e.g., MAP, in order to make the results comparable across different climate and vegetation zones. In addition, we consider two measures of intra- and inter-annual rainfall variability: (2) the Walsh-Lawler seasonality index[32], which measures how precipitation is distributed throughout the year—from low (precipitation is similar

between months) to high (annual precipitation is concentrated in a short period) (Fig. 1E); and (3) the year-to-year variability of rainfall, which we define as the normalized standard deviation of annual precipitation (AP) sums. As total MAP and the standard deviation of AP are −as should be expected−highly correlated (Supplementary Fig. S2), we normalize the standard deviation of AP by the MAP pixelwise, giving a suitable normalized inter-annual precipitation variability estimate (Fig. 1F). Further, we consider a reanalysis-based soil moisture estimate as an additional proxy for plant-available water[33].

To measure resilience, we rely here on a direct empirical quantification of resilience in terms of the recovery rate from large perturbations[4], as well as two different theory-based estimates of the restoring rate $\lambda$, derived from the AC1 and from the variance[4]. We note that we define the recovery rate $\lambda$ as a negative number; values closer to zero imply slower recovery and hence lower resilience; correspondingly, higher AC1 and variance values imply lower resilience (see Methods).

### Joint effects of water availability and precipitation variability on vegetation resilience globally

To assess the first-order relationships between vegetation resilience and water availability, we first consider all land-cover types and climate zones together (Fig. 2). For all three vegetation indices (VOD and two NDVI datasets), we find that the highest recovery rates are generally found in areas of water surplus (Fig. 2, left of dashed line).

The relationship to shorter-term precipitation variability is less clear, however. Intra-annual precipitation variability does not scale cleanly with recovery rate (Fig. 2, left column); there exist highly resilient areas which receive precipitation only during short time periods. These areas are found exclusively in grass and shrublands globally and are concentrated in the African Sahel, where plants are adapted to

highly seasonal precipitation. In contrast, high inter-annual precipitation variability (Fig. 2, right column) leads to almost universally lower resilience, indicating that more consistent precipitation year-on-year encourages more resilient vegetation. At the global scale, we thus infer a clear increase in vegetation resilience with increasing water availability and with decreasing inter-annual precipitation variability (Fig. 2B, D, F); the relationship between resilience and precipitation seasonality is less concise (Fig. 2A, C, E).

### Long-term water availability

The revealed concise relationship between aridity and resilience (Fig. 2) is broadly consistent across most land-cover types (Fig. 3). Resilience tends to increase non-linearly with increasing water availability; some land covers show distinctly stronger aridity/AC1-derived $\lambda$ relationships as landscapes transition from water-balance (aridity ~ 1) to water deficit (aridity >1). For example, Savannas (olive line, Fig. 3) show a sharp transition at aridity ~ 1 for MODIS NDVI (the linear slope for aridity <1 vs empirical $\lambda$ (AC1 $\lambda$) is 0.26 (0.18), compared to 0.13 (0.05) for aridity >1). For both VOD and NDVI, vegetation resilience changes in many land-cover types plateau above aridity >2, which roughly demarcates the transition into semi-arid environments[31]. In these regions, grass and savanna landscapes dominate (Fig. 3); it is likely that plant adaptations to water limitations[34] account for some of this asymptotic behavior.

The relationships between aridity and resilience are overall consistent for the NDVI and VOD data, but not identical. While it is not possible to identify a single cause across all ecosystems, differences in the attributes measured by NDVI and VOD, respectively—and the intrinsic internal variability of each dataset—likely drive the heterogeneity. It should be noted that NDVI reflects vegetation chlorophyll content or photosynthetic activity, whereas VOD reflects vegetation density and productivity. While some differences regarding the estimated resilience should therefore be expected, the overall similarity between the results obtained for the three data sources provides a strong argument that the inferred reduction of resilience with higher aridity across land-cover types is robust. In particular, the fact that we obtain similar results for the single-sensor MODIS NDVI as for the other two, mixed-sensor datasets, implies that the merging of signals for the latter data products do not affect our results.

### Intra- and inter-annual precipitation variability

While there is a clear demarcation between the resilience of vegetation in water-surplus and water-deficit regions at the global scale (Fig. 2) and when separated by land cover (Fig. 3), the role of intra-annual water variability is not as clear (Fig. 4). We find that resilience is broadly similar across precipitation seasonalities; however, Kendall-Tau coefficients remain generally positive, implying decreasing resilience with more seasonal precipitation. Relationships vary across land-cover types; grass-dominated landscapes in particular (woody savanna, savanna, grasslands) have regions of both positive and negative relationships between resilience and seasonality. We posit this is due to the wide distribution of these land-cover zones across the globe (Fig. 1C)—woody savannas are dominant in both central Africa and at high northern latitudes in Canada and Siberia.

Vegetation is not only sensitive to the distribution of precipitation within the year, but also to its distribution between years[30,35]. Across all three datasets and measures of resilience, we find that higher relative inter-annual variability of precipitation leads to less resilient vegetation, particularly in grass-dominated landscapes (Fig. 5, Supplementary Fig. S8). We further find that inter-annual precipitation variability (Fig. 5) is a relatively stronger control on resilience than intra-annual precipitation distribution (Fig. 4). We posit that this difference is due to the characteristic time scales at which vegetation responds to water deficits, with longer (inter-annual) time scales being relatively harder to adapt to. In regions with highly seasonal precipitation, vegetated

ecosystems have adapted to annual and short-term water deficits with a variety of methods (for example, phenotypic plasticity, drought pruning)[34,36,37]. In contrast, inter-annual and longer-term water deficits can cause large shifts in vegetation (and ecosystem) species mixes[36]. It is also important to note that inter-annual precipitation variability is not static; recent research has found spatially heterogeneous changes in water deficits (i.e., drought events)[9,35,38]. Our results indicate that increasingly frequent and extreme water deficits—especially those at the multi-annual scale—will impact the resilience of vegetation ecosystems worldwide. Such events will have a relatively larger impact than intra-annual precipitation variability on vegetation resilience as ecosystems lose the ability to recover to their previous state.

It should be noted that we do not see a decrease in vegetation resilience when considering the inter-annual precipitation variability without normalizing by the total annual precipitation sums (Supplementary Fig. S10); higher absolute precipitation variability estimates correspond to higher vegetation resilience. This positive correlation is driven primarily by the absolute MAP itself; the standard deviation of annual precipitation sums is higher in regions that have overall higher MAP, since it cannot be negative. If we limit our analysis to only a small precipitation range around the median (40–60th percentile of annual precipitation values by land cover) to account for this in an alternative way to normalizing, the pattern is the same as for the normalized inter-annual precipitation variability. Namely, we find lower resilience for higher precipitation variability (Supplementary Fig. S11), even if we do not normalize inter-annual precipitation variability by MAP (Supplementary Fig. S12).

## Discussion

Water availability plays a primary role in controlling the occurrence, type, and health of vegetation globally. Our work extends that of previous research[1], and documents a global relationship of less resilient vegetation with lower water availability (Fig. 1), lower MAP (Supplementary Fig. S4), and lower soil moisture (Supplementary Fig. S5). This relationship is consistent across land-cover types, with many land covers showing the greatest changes in vegetation resilience as aridity approaches 1 (i.e., a balance between precipitation and potential evapotranspiration) (Figs. 2, 3).

The total yearly amount of precipitation, however, is not the only control on the health and resilience of vegetation. We find a globally consistent pattern—particularly in grass-dominated regions—of decreasing vegetation resilience with higher inter-annual precipitation variability (Fig. 5). While more work is required to fully constrain the mechanisms behind this response, we posit that vegetation reliant on surface water and direct precipitation is more strongly impacted during low-precipitation periods than tree-dominated areas with deeper root systems and access to longer-term water storage (e.g., lakes, rivers, and shallow groundwater). It has previously been shown that vegetation productivity is tightly coupled to antecedent precipitation in arid to semi-arid environments[30] and that regions with higher woody biomass are relatively buffered against short-term water deficits[29,35]. Our results confirm these observations at the global scale, and establish an additional link between precipitation variability and vegetation resilience.

An important caveat is that we do not examine long-term changes in precipitation or climatic conditions. All else being equal, changes in precipitation will engender a change in vegetation, which generally will be expressed in our resilience metrics. We aim in this study to map long-term global patterns; our analysis framework here does not allow us to disentangle whether vegetation resilience has changed[4], and to what degree those changes are driven by precipitation changes.

We further note that differences in resilience between land-cover types likely reflect mainly intrinsic, physiological differences in vegetation—with all other drivers fixed, vegetation will grow more quickly in a rainforest than in a savanna. The measured differences also

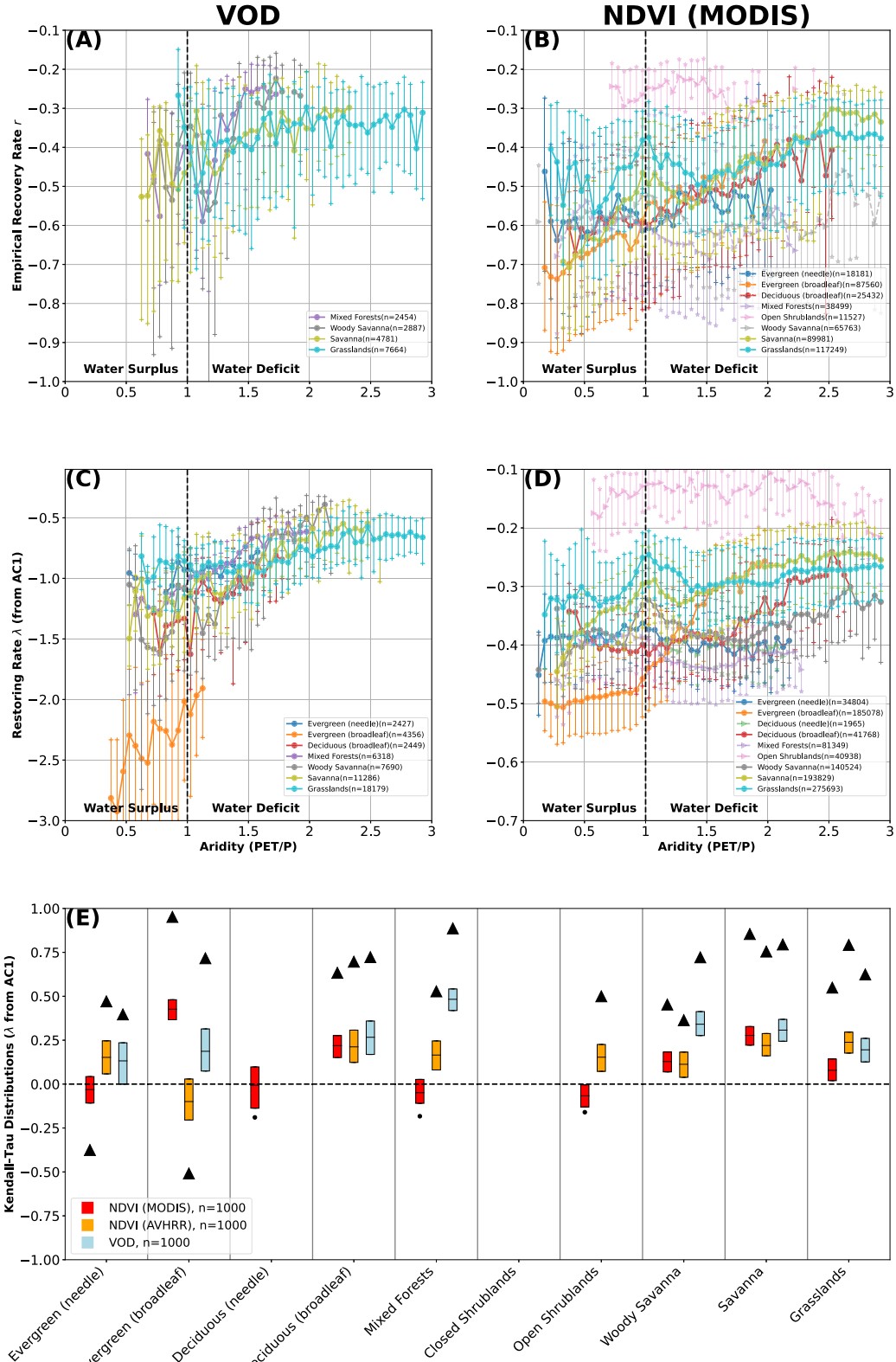

depend on how well satellite data products capture fine-scale changes in vegetation of different densities and structures. Our discussion hence focuses on the relationship between vegetation resilience and water availability within single land-cover classes. Despite this control, however, there remain large differences in vegetation within the same broad land-cover classes at the global scale. Our results (Figs. 2–5) thus include an array of different vegetation mixes and responses to

precipitation variability; further and smaller-scale work would be required to constrain how different plant responses to precipitation variability impact their resilience to changing environmental conditions.

Despite considerable spatial heterogeneity[39], a large body of work points to increasing precipitation variability in the coming decades in response to anthropogenic climate change[9]. The global tendency is for

**Fig. 3 | Vegetation resilience as a function of aridity[31] at the global scale, separated by land-cover type[46].** Vegetation resilience $\lambda$ estimated empirically (**A**, **B**) and via the AC1 (**C**, **D**) for vegetation optical depth (VOD, left column) and MODIS NDVI (right column). Binned medians shown as solid dots (Kendall-Tau (KT) $p < 0.05$) and transparent arrows (KT $p > 0.05$), with 25–75th percentiles of each bin shown as connected vertical lines capped with hatches. Land covers with less than 1000 points or less than 10 bins of at least 50 members are omitted. **E** KT coefficients (aridity vs AC1-derived $\lambda$, panels **C**, **D**) for each land-cover type. Significant ($p < 0.05$) KTs shown as a black triangle (KT of median binned data, cf. **C**, **D**), insignificant relationships ($p > 0.05$) shown as a black circle. Additional box-plot of 1000 randomly sampled surrogates (box edges: 25–75th percentiles, black line: median) shown with red for MODIS NDVI, orange for AVHRR NDVI[24], and blue for VOD. KT of medians consistently higher than box plots due to random sampling (see Methods). Both VOD and NDVI exhibit lower resilience—i.e., $\lambda$ closer to zero, see Methods—with lower water availability across the majority of land-cover types. Equivalent figure for mean annual precipitation (MAP) shown as Supplementary Fig. S4, and for mean annual soil moisture shown as Supplementary Fig. S5. Figure for aridity showing all three instruments and metrics as Supplementary Fig. S6.

wetter regions to receive more water and drier regions to become drier, as well as for increases in both wet and dry extremes[9,35,40]. Based on our research, it is clear that changes in both precipitation volume and variability will have a measurable and spatially heterogeneous impact on global vegetation (Figs. 2–5). Low- and variable-precipitation regions will face comparably higher burdens in response to increasing precipitation variability in a warming world; this shift has strong implications for future ecosystem functioning in vast parts of the sub- and extra-tropics. Resilience loss, and eventually potential desertification of grass- and shrublands, could trigger a chain of destabilizing feedbacks; for example, vegetation resilience loss and ecosystem transitions could reduce water storage capabilities at continental scales[41], affect rainfall patterns due to atmosphere-vegetation interactions[8], and accelerate greenhouse gas emissions[11,42].

Our analysis rests on both empirical recovery rates from perturbations – direct estimates of vegetation resilience according to the common definition—and two theory-based estimates (from AC1 and variance) that are hence more indirect. Although our results are globally coherent, we therefore note that especially our two theory-based resilience estimates may, in principle, be influenced by local-scale factors. For example, the varying physiological characteristics of different vegetation types contained within each time series will present a mixed signal in many regions. We have minimized this influence by performing our analyses separately for different land covers, and by limiting our work to natural land covers. However, it is possible that some regions retain mixed signals (cf. Fig. 4), which could influence local-scale resilience estimates. At the global scale, however, we find overall consistent behavior across climate and vegetation zones. Using VOD at the global scale, we have previously empirically confirmed the theoretical relationships between AC1 and variance on the one hand, and empirically inferred recovery rates on the other hand, suggesting that the AC1 and variance can indeed serve as resilience metrics[4]. The consistency between our results for the empirical recovery rates and the theory-based estimates across land-cover types adds strong further independent evidence to this. In particular, the fact that we find very similar results for the empirical recovery rates and the theory-based estimates for the MODIS NDVI data show that this confirmation is not impacted by merging data from different sensors.

We have presented evidence based on both empirically estimated recovery rates and different theoretical—yet empirically confirmed—resilience metrics for concise global relationships between vegetation resilience and water availability, modulated by land-cover type. We find overall greater resilience in regions with higher water availability across climate zones and vegetation types, based on an aridity index. However, our results also suggest that resilience consistently declines with increasing precipitation variability especially on inter-annual time scales, and particularly in grass-dominated landscapes. Simulations from the sixth phase of the Climate Model Intercomparison Project suggest increased precipitation variability under global warming scenarios in the coming decades. Based on our empirical results we hence infer an increasing risk of vegetation degradation and eventually desertification—especially in regions with savanna, grass- and shrublands—in response to anthropogenic climate change.

## Methods

### Vegetation and land-cover data

To monitor vegetation at the global scale, we use three datasets: (1) vegetation optical depth (VOD, 0.25°, Ku-Band, daily 1987–2017[23]) (Fig. 1A), (2) AVHRR GIMMSv3g normalized difference vegetation index (NDVI, 1/12°, bi-weekly 1981–2015[24]) (Fig. 1B), and (3) MODIS MOD13 NDVI at 0.05° (16-day, 2000–2021[25]). We correct for spurious values in the NDVI data (e.g., cloud contamination) using the method of Chen et al.[43]. We resample the VOD data using bi-weekly medians to agree with the NDVI data time sampling.

For all three vegetation datasets, we remove seasonality and long-term trends using seasonal trend decomposition by Loess[4,44] based on the proposed optimal parameters listed in Cleveland et al.[44] (code available on Zenodo[45]). That is, we use a period of 24 (bi-monthly, 1 year), 47 for the trend smoother (just under 2 years) and 25 for low-pass (just over 1 year). We only use the STL residual—the de-seasoned and de-trended NDVI and VOD time series—in our analysis.

To contextualize our understanding of vegetation resilience, we use MODIS MCD12Q1 land cover[46] (Fig. 1C) as well as a global average aridity index based on WorldCLIM data[31] (Fig. 1D). We exclude from our analysis anthropogenic and non-vegetated landscapes (e.g., permanent snow and ice, desert, urban), as well as any land covers which have changed (e.g., forest to grassland) during the period 2001–2020.

### Precipitation data and variability metrics

To measure precipitation at the global scale, we rely upon ERA5 data (~30 km, monthly, 1981–2021)[33]. We process global-scale precipitation metrics using the Google Earth Engine[47] platform. We further use the sum of soil moisture from the surface down to 28 cm of depth (first two layers of the ECMWF Integrated Forecasting System soil moisture estimates) to quantify soil moisture means and inter-annual variability[33].

It is well-documented that vegetation resilience is responsive to the MAP of certain regions[1]. However, the role of precipitation variability in controlling vegetation resilience has not been well-studied. Here we examine precipitation variability in terms of both intra- and inter-annual patterns. Intra-annual precipitation variability is determined in terms of the Walsh-Lawler Seasonality index[32] (Fig. 1D), calculated using monthly data from ERA5[33].

Partly due to the fact that precipitation is non-negative, simple inter-annual variability metrics such as the standard deviation of annual precipitation sums are biased by the absolute precipitation sums; higher precipitation regions have a higher possible range of variability. To limit the influence of MAP, we hence investigate the standard deviation of annual precipitation sums normalized by the MAP, over the period 1981–2021, based on ERA5 data[33] (Fig. 1F). We motivate our normalization by MAP with the strong linear relationship between MAP and MAP standard deviation (Supplementary Fig. S2). We further confirm our discovered relationships (Fig. 5) using only those regions where MAP was between the 40 and 60th percentile of MAP for a given land cover (Supplementary Figs. S11,S12). This serves as an additional check that our normalization of MAP standard deviation by MAP does not bias the inferred relationship between vegetation resilience and precipitation variability. Similarly, we generate a normalized inter-annual soil moisture

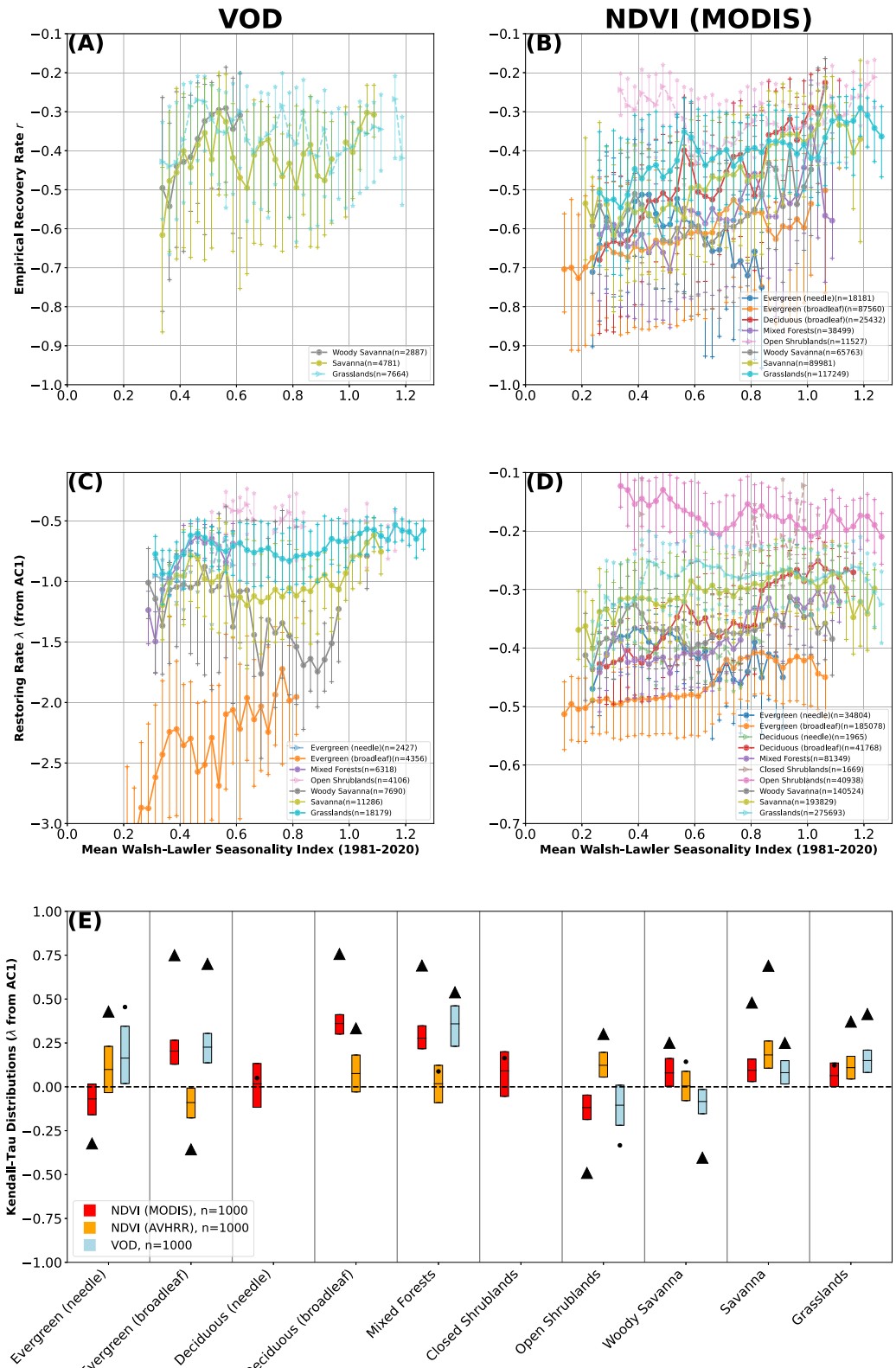

variability by normalizing year-on-year soil moisture standard deviation (Supplementary Fig. S8) by long-term mean soil moisture (Supplementary Fig. S5).

## Empirical resilience estimation

Resilience is defined as the ability of a system to recover from perturbations, and can be quantified empirically by the speed of recovery

to the previous state[16,17]. To measure resilience on the global scale, we employ a recently introduced methodology[4] which we will briefly summarize in the following.

We first identify sharp transitions in the vegetation time series using an 18-point (9 month) moving window to define local slopes throughout the time series[48]. We then identify slopes above the 99th percentile, and define connected regions as individual perturbations.

**Fig. 4 | Vegetation resilience as a function of precipitation seasonality in terms of the Walsh-Lawler seasonality index[32] (Methods), separated by land-cover type[46].** Vegetation resilience $\lambda$ estimated empirically (**A**, **B**) and via the AC1 (**C**, **D**) for vegetation optical depth (VOD, left column) and MODIS NDVI (right column). Binned medians shown as solid dots (Kendall-Tau (KT) $p < 0.05$) and transparent arrows (KT $p > 0.05$), with 25–75th percentiles of each bin shown as connected vertical lines capped with hatches. Land covers with less than 1000 points or less than 10 bins of at least 50 members are omitted. **E** KT coefficients (aridity vs AC1-derived $\lambda$, panels **C**, **D**) for each land-cover type. Significant ($p < 0.05$) KTs shown as a black triangle (KT of median binned data, cf. **C**, **D**), insignificant relationships

($p > 0.05$) shown as a black circle. Additional box-plot of 1000 randomly sampled surrogates (box edges: 25–75th percentiles, black line: median) shown with red for MODIS NDVI, orange for AVHRR NDVI[24], and blue for VOD. KT of medians consistently higher than box plots due to random sampling (see Methods). Both VOD and NDVI exhibit lower resilience−i.e. $\lambda$ closer to zero, see Methods−with lower water availability across the majority of land-cover types. While for all three considered vegetation datasets empirical recovery rates generally decrease with more concentrated precipitation, the relationship between Walsh-Lawler seasonality and recovery rates is less steep than for aridity (Fig. 3). Figure showing all three instruments and metrics as Supplementary Fig. S7.

The highest peak (largest instantaneous slope) within each connected region is then labeled as an individual disturbance.

The employed approach does not delineate every rapid transition in a time series due to our reliance on percentiles; our dataset will be inherently biased towards the largest transitions. Furthermore, the same transitions are not guaranteed to be captured for both NDVI and VOD data in each location, as the percentiles will naturally vary between the datasets. Finally, our method will in some cases produce false positives, especially in cases where a given time series does not have any significant rapid transitions. To limit the influence of false positives on our results, we discard any perturbations where the time series does not drop significantly, and where the period before and after a given transition does not pass a two-sample Kolmogorov–Smirnov test[4].

Finally, using our global set of time-series transitions, we can identify each local vegetation (NDVI or VOD) minima, and use the five following years of data to fit an exponential function to the residual time series, assuming that the recovery after a perturbation to a vegetation state $x_0$ follows approximately the equation

$$x(t) \approx x_0 e^{rt} \tag{1}$$

where $x(t)$ denotes the vegetation state at time $t$ after the perturbation. Negative $r$ indicates that the vegetation system will return to the original stable state at rate $|r|$. For positive $r$, the initial perturbation would be amplified, suggesting a non-resilient vegetation state. Our empirical recovery rates are defined as the fitted exponent $r$, obtained for each detected transition in the NDVI and VOD residual time series. We finally use the coefficient of determination $R^2$ to remove instances where the fitted exponential poorly matches the underlying data[4].

For the empirical estimate of the restoring rate obtained from fitting an exponential to the recovery after an abrupt negative deviation of VOD or NDVI, abrupt changes in the mean state induced by changing sensors rather than an actual vegetation shift may impact the results. However, all datasets used here are tightly cross-calibrated to eliminate mean-shifts when new instruments are introduced[23,24]. It is therefore unlikely that changes in the instrumentation of the various datasets unduly influence our empirical estimates of $\lambda$.

## Dynamical system metrics of resilience
The lag-one autocorrelation (AC1) has previously been proposed to measure the stability of real-world dynamical systems in general, and the resilience of vegetation systems in particular[1,19–21,49]. Based on the concept of critical slowing down, the AC1 has, together with the variance, also been suggested as an early-warning indicator for forthcoming critical transitions[50,51]. Mathematically, the suitability of the variance and AC1 as resilience measures and early-warning indicators can be motivated as follows[4,52,53]. First, linearize the system around a given stable state $x^*$:

$$d\bar{x} = \lambda \bar{x} dt + \sigma dW \tag{2}$$

for $\bar{x} := x - x^*$, assuming a Wiener Process $W$ with standard deviation $\sigma$. The dynamics are stable for $\lambda < 0$ and unstable otherwise. Upon

discretizing the resulting Ornstein-Uhlenbeck process into time steps of width $\Delta t$, the variance and AC1 of the resulting order-one autoregressive process are then related to the restoring rate $\lambda$ via[54]:

$$\langle \bar{x}^2 \rangle = -\frac{\sigma^2}{2\lambda} \tag{3}$$

for the variance, and

$$\alpha(n) = e^{n\lambda\Delta t} \tag{4}$$

for the AC1. Hence, the closer $\lambda$ is to zero, the larger the AC1 and variance, corresponding to lower stability. Note that theory[55] suggests that the recovery rate $r$ is equal to the restoring rate $\lambda$. An empirical global confirmation for this relationship for has recently been demonstrated based on both NDVI and VOD data[4]. We compare the dependence of the recovery rate $r$ and two different theoretical estimates of the restoring rate $\lambda$−obtained via inverting the above equations for the variance and AC1−with respect to their dependence on water availability and its variability.

It is important to note that combining data from different sensors with varying signal-to-noise ratios (e.g., VOD, AVHRR NDVI) can bias estimates of temporal changes in resilience indicators because the higher-order statistics of the resulting time series are not homogeneous[4,26]. In the present work, however, we do not investigate temporal trends via estimating resilience indicators in sliding windows (as in refs. [4,26]), but rather estimate resilience indicators for the full available time series. This excludes the possibility of systematic biases in our AC1- and variance-based estimates of the restoring rate $\lambda$. In principle, the combination of different sensors might lead to larger uncertainties in the estimates of $\lambda$: combining different sensor data leads to temporally varying (yet spatially homogeneous) effects on the AC1 and variance and may therefore lead to a wider spread−but not a bias−in the resulting estimates of $\lambda$.

## Binning and significance testing
The direction and magnitude of our discovered relationships (e.g., Figs. 3–5, Supplementary Figs. S4–S12) will be to some degree controlled by the choice of bin sizes. We tested three bin sizes for each different variable (Supplementary Figs. S13,S14). We also imposed the conditions that there were at least 50 measurements in each bin to form a proper median, and that we only report those relationships which cover ten or more bins.

To better constrain the relationship between each driving variable and resilience, we use the non-parametric Kendall-Tau test[56]. Kendall-Tau statistics are calculated over each set of binned medians, as well as using a Monte-Carlo approach. Over 1000 iterations, we choose one random point from each bin and recalculate the Kendall-Tau statistics. These 1000 surrogates are displayed as box plots in Figs. 3–5. Note that the Kendall-Tau value of the median line will almost always be larger than the median of the 1000 Kendall-Tau values resulting from the surrogates due to the smoothing inherent in taking binned medians. That is, the binned medians represent a smooth and almost monotonic line with fewer jumps, while the 1000 surrogates will have strong fluctuations from one bin to the

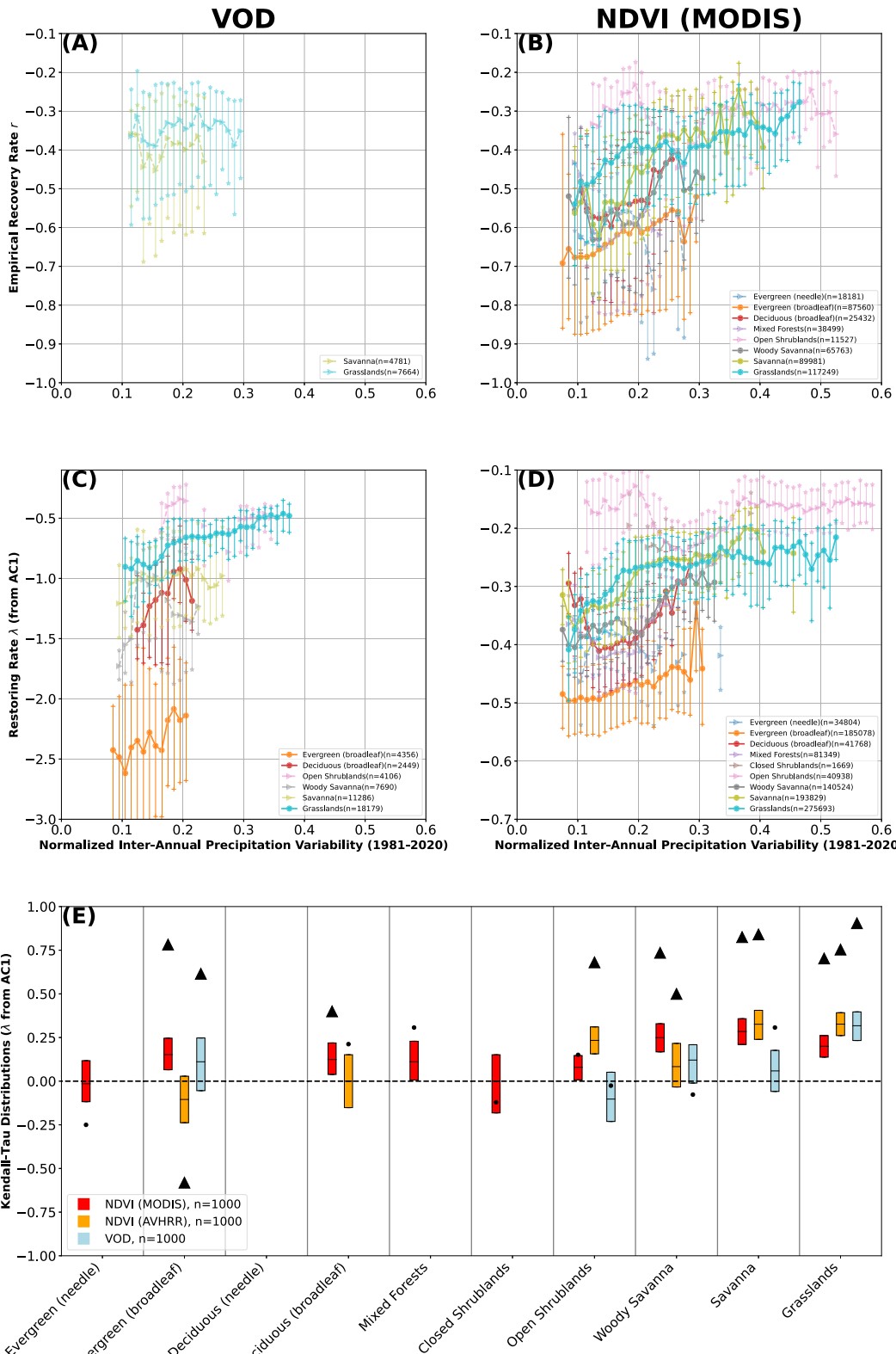

next, leading to overall lower Kendall-Tau values. The fraction of Kendall-Tau statistics which share a sign with the Kendall-Tau of the median line is also reported on Supplementary Figs. S4–S14.

We find that our Kendall-Tau statistics are robust against changing bin sizes (Supplementary Figs. S13,S14), where the direction of trends does not change from those reported in Fig. 3. The magnitude of the Kendall-Tau statistic—as well as $p$-values—shift with different bin sizes,

with smaller bin sizes typically resulting in more robust trends. Changes in bin size do not have a strong impact upon our data interpretations or conclusions.

**Reporting summary**
Further information on research design is available in the Nature Portfolio Reporting Summary linked to this article.

**Fig. 5 | Vegetation resilience as a function of normalized inter-annual precipitation variability (Methods), separated by land-cover type[46].** Vegetation resilience $\lambda$ estimated empirically (**A**, **B**) and via the AC1 (**C**, **D**) for vegetation optical depth (VOD, left column) and MODIS NDVI (right column). Binned medians shown as solid dots (Kendall-Tau (KT) $p < 0.05$) and transparent arrows (KT $p > 0.05$), with 25–75th percentiles of each bin shown as connected vertical lines capped with hatches. Land covers with less than 1000 points or less than 10 bins of at least 50 members are omitted. **E** KT coefficients (aridity vs AC1-derived $\lambda$, panels **C**, **D**) for each land-cover type. Significant ($p < 0.05$) KTs shown as a black triangle (KT of

median binned data, cf. **C**, **D**), insignificant relationships ($p>0.05$) shown as a black circle. Additional box-plot of 1000 randomly sampled surrogates (box edges: 25–75th percentiles, black line: median) shown with red for MODIS NDVI, orange for AVHRR NDVI[24], and blue for VOD. KT of medians consistently higher than box plots due to random sampling (see Methods). For both VOD and NDVI we infer lower resilience for higher relative inter-annual precipitation variability. Equivalent figure for normalized inter-annual soil moisture variability shown as Supplementary Fig. S8. Figure showing all three instruments and metrics as Supplementary Fig. S9.

## Data availability
The Kendall-Tau statistics generated in this study have been deposited on Zenodo at 10.5281/zenodo.7436669[45]. The raw environmental and satellite data used in this study are publicly available[23–25,31,46]. Direct links to the datasets can be found at: GIMMS NDVI: https://www.cen.uni-hamburg.de/en/icdc/data/land/gimms-ndvi3g.html, VOD: https://zenodo.org/record/2575599, MODIS Land Cover: https://lpdaac.usgs.gov/products/mcd12q1v006/, MODIS NDVI: https://lpdaac.usgs.gov/products/mod13c1v006/, WorldCLIM Aridity: https://figshare.com/articles/dataset/Global_Aridity_Index_and_Potential_Evapotranspiration_ET0_Climate_Database_v2/7504448/4, ERA5 Climate Data: https://www.ecmwf.int/en/forecasts/datasets/reanalysis-datasets/era5.

## Code availability
The code used for de-seasoning the data via STL, detecting abrupt transitions, and estimating resilience both empirically and via time-series metrics (AC1, Variance) can be found on Zenodo: https://doi.org/10.5281/zenodo.7436669[45]. Code is written in Python (3.9.13).

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

## Acknowledgements
The State of Brandenburg (Germany) through the Ministry of Science and Education and the NEXUS project supported T.S. for part of this study. T.S. also acknowledges support from the BMBF ORYCS project and the Universität Potsdam Remote Sensing computational cluster. N.B. acknowledges funding by the Volkswagen foundation. This is TiPES contribution #208; the TiPES ('Tipping Points in the Earth System') project has received funding from the European Union's Horizon 2020 research and innovation program under grant agreement No. 820970. N.B. acknowledges further funding by the European Union's Horizon 2020 research and innovation program under the Marie Sklodowska-Curie grant agreement No. 956170, as well as from the Federal Ministry of Education and Research under grant No. 01LS2001A. Open access publication funded by the Deutsche Forschungsgemeinschaft (DFG, German Research Foundation) – Projektnummer 491466077.

## Author contributions
T.S. and N.B. conceived and designed the study. T.S. processed the data and performed the numerical analysis. Both authors interpreted the results and wrote the manuscript.

## Funding

## Competing interests
The authors declare no competing interests.
