## [Peer Review File · Nature Communications]

Reviewer comments, first round -

Reviewer #1 (Remarks to the Author):

The submitted paper by Smith and Boers evaluates the 'resilience' of different land cover types globally. Vegetation resilience has many definitions throughout the literature, and here the authors use timeseries of VOD and NDVI to determine the rates at which ecosystems return a de-trended pre-disturbance mean state. The authors present four different ways to calculate the same metric, their 'recovery rate', and how this metric varies with aridity (improperly defined) and precipitation variability. A few things that should be addressed:

While the metric itself, and how it is calculated, are interesting and novel, much of its development as a new approach as well as the global patterns of this new metric are already in review in another paper that is included here. Many studies have examined vegetation resilience through other metrics, and a better case why this new metric is needed should be made. As given in the methods (which is missing some definitions for symbols used) the 'recovery rate' should be proportional to the natural log of the autocorrelation, which. More effort needs to be made justifying why another metric, that scales with ones presented in other studies, should be studied.

It seems to me that the main significance of this study is relating the authors new metric to precipitation and aridity. For this, based on the current presentation, it is not clear how much of a control the climatic drivers exert relative to the vegetation itself. For example, does the precipitation have a similar 'recovery rate' as the vegetation? I.e., is what is being reported the recovery of the climate or of the vegetation. These may not be able to be disentangled, but the point needs some clarification.

As a reader of this study, some of the choices that the authors have made are unclear to me. Details as to why VOD or LAI are both used are not elaborated well. Why present both, does one present more, or different information than the other? The same can be said about the different precipitation metrics. Similarly, while the authors claim to have 4 different metrics of resilience, they all can be derived, or result in, the same conceptual model of exponential recovery of vegetated ecosystem post disturbance. In particular, for the 3 models derived from the assumption of an Ornstein-Uhlenbeck process, which of these three best captures this metric, are differences expected between the three? If something truly adheres to this type of process (which can be simulated easily) which of these three is expected to give the best estimate of the true parameter?

Finally, while of minor import to the conclusions in this paper, the Aridity Index provided by WorldCLIM (Trabucco & Zomer 2019) is the inverse from what is commonly understood in hydrology and climate science, where this quantity is typically calculated as PET/P not P/PET. Aridity is defined as PET/P because it is a measure of how arid a location is, with increasing values corresponding to more arid locations. P/PET is a wetness index. Aridity defined as PET/P is consistent with Bydtko's framework delineating water and energy limits for terrestrial systems, though how to specify PET is debated. See Greve et. al. and many of the papers within (DOI: 10.1088/1748-9326/ab5046) for more details. Please clarify this point in the present study.

A few other minor considerations:

- Consider expressing the recovery rate as a time value instead of a rate. The inverse of the recovery rate could easily be converted into recovery rate half-life and presented in units of months or years. This might help readers more easily understand the presented results.
- Be very clear that you're using the inverted definition of aridity. Explicitly state P/PET in the figure legend
- More details on the sample sizes, resolutions in space and time, about the data products and the confidence in the derived metrics need to be given.
- Rainforests are not water limited but energy limited, so more rainfall leads to less productive systems, this might explain some of your anomalous results for very wet systems
- No consideration for the interaction between precipitation and aridity is presented and might be

worth considering. These are not independent variables. Wet and dry systems might react differently at different levels of precipitation variability.

- If there is not clear reason to choose one of these metrics over another, then would a multi-metric average be more reliable?

Overall, I think the study and its approach are interesting and do not see any major flaws in the methodology. More details are needed to help make the case for the importance of this study, how it separates itself from prior work, how the choices of metrics and their calculation were decided, and how the observed patterns relate to overarching temporal patterns in climate.

- Stephen Good

Reviewer #2 (Remarks to the Author):

The authors employ both theoretical and empirical resilience indices to quantify how precipitation availability and variability impact vegetation resilience at a global scale. This is an interesting study that shows an increasing resilience of vegetation with aridity across different ecosystems. By contrast, the grasslands showed decreased vegetation resilience under high interannual precipitation variability. This finding is important because the increasing precipitation variability will lead to critical challenges for the state and stability of vegetation systems. The manuscript is clearly written, and the key message is nicely illustrated. However, the evidence (or results) to support the conclusions is simple and weak, and it is too subjective to interpret the results in this manuscript. I have the following comments for the authors to consider:

1. The authors quantify resilience as a function of aridity and precipitation interannual variability. How about the relationships between interannual variability parameters and aridity? Whether the contrasting effects of precipitation availability and variability results from the negative correlations between precipitation availability and variability? For instance, in the grasslands, the annual precipitation is lower and the relative deviation from precipitation is generally higher. In comparison, the precipitation is high and evenly distributed over the years in the tropics. In other words, whether they do necessarily show a contrasting behavior?

2. The title, the abstract, and the study objective all clearly state that the interest here is how key dimensions of precipitation regimes affect vegetation resilience. My concern is that in the manuscript, effects in precipitation amount and interannual variability dominate the results, but precipitation variability includes many other components as well.

3. The authors believe the similar pattern shown in Figure S3 & S6 and Figure 2 stem from the spatial coherence between mean annual precipitation and the three components of precipitation variability (i.e., the standard deviation of annual precipitation, annual precipitation frequency, and seasonality). All existing precipitation metrics are correlated to some extent. However, these metrics represent different mechanisms (drought, seasonality, and precipitation irregularity). In my opinion, interannual variability alone does not retain the full spectrum of information on precipitation variability. Why not use relative variability estimates to describe other aspects of precipitation variability? How vegetation resilience is affected by intra-annual variance of precipitation?

4. Figure 2 & 3: a robust statistic is required to quantify the response of vegetation resilience to precipitation availability and variability instead of describing the resilience trend only. The authors should stress out the statistical robustness of this study in the method section. For example, the error bars shown in Figure 2 are very long, so I am curious that whether the correlations are significant for most land cover types. In addition, all correlations in this manuscript only presented crude trends (e.g., decrease or increase), there are no fitted equations, R²-values, and even P-values were provided. Therefore, it is unconvincing, in terms of me, on assessing the correlations between vegetation resilience and precipitation availability and variability.

5. All the results seem to rely on the binning analysis but the methods for this binning issue are not included in the study Methods section. The precipitation bin used in this study might bias the

trend of vegetation resilience. I would like to see sensitivity tests to the bin size. Whether bin size change the trend of raw data? or whether each bin had enough samples?

6. The descriptions on correlations between vegetation resilience and precipitation availability and variability are so simple, e.g. only decrease or increase. The correlation between restoring rate λ and aridity (Figure 2), for example, presents remarkable non-linear regression, e.g., keeping relative stable after about 0.6-0.75 (visual estimate), which is near the threshold that distinguishes the Semi-arid and Dry sub-humid (or Humid) (Trabucco and Robert 2018, UNEP 1997). If we considered the threshold, the results might correspond to another different story.

7. It is unbelievable that the authors explain the inconsistent patterns in evergreen broadleaf forests by "data unreliable" in lines 82-83. If the data of evergreen broadleaf forests are unreliable, how to prove that the data for other land cover types are reliable. Therefore, it is inadvisable to deal with unexpected results in this way. This interpretation gives me a sense that it doesn't matter what the results are, we would or should only keep the conclusions are consistent with common sense.

8. Line 154-161: The aspect of plasticity is not discussed in the paper. There are many mechanisms for plants to adjust to changing environmental conditions and lower vegetation resilience of grass-dominated regions may also reflect a structural adjustment to higher interannual precipitation variability.

Other comments:

1. The authors select both NDVI and VOD data to quantify vegetation resilience patterns. Why the spatial distribution of VOD is presented in Figure 1 but keep the NDVI in Figure S1?
2. The source of precipitation data should also be included in the Methods.
3. Figure 2, 3 and everywhere: Please give the results of linear regressions between aridity and λ . The error bars shown in Figure 2 are very long, so I am curious that whether the correlations are significant for most land cover types?
4. Line 146: Supplemental Figure S3?

Reviewer #3 (Remarks to the Author):

This study looks at the resilience of global ecosystems as a function of aridity and precipitation (variability). The topic is of high relevance and the methodology (apart from the empirical method submitted elsewhere by the lead author) well established. There is novelty in the fact that apart from assessing resilience worldwide and using times series of AVHRR NDVI, the study uses a new, long-term vegetation optical depth dataset from passive microwave observations.

Even though the study is performed in a sound way and very well written, I have strong concerns about the conclusions drawn from it. I am even more worried about the conclusions drawn by the study submitted by the authors elsewhere, i.e.: "Smith, T., Boers, N., & Traxl, D. (in review). Satellite-based evidence for global loss of vegetation resilience.", which unfortunately is not publicly accessible.

The issue is that looking at such delicate statistical metrics from long-term observations assumes that these datasets have homogeneous characteristics in space and time. As correctly noticed by the authors, NDVI is known to saturate for denser canopies. But even for microwave observations, which have much longer wavelengths, this may be the case. Typically, the signal-to-noise ratios of remotely-sensed vegetation datasets are optimal at intermediate vegetation densities, when confounding background (soil) effects are relatively low but the observations do not yet saturate. These are vegetation types where strongest response was found. Also in the temporal domain, retrieval skills are not static with newer satellites typically being less noisy. But, more importantly, merged datasets like the VODCA product used in this study, have a strongly variable number of observations used per satellite-blending period, which has a direct impact on the noise, and hence metrics like AR1.

As mentioned above, I am particularly worried that the "loss-of-vegetation-resilience signal" seen in the paper that is in review is simply a result of the increased product skill over time.

In general, the quality of the current paper can only be fully assessed once details on the "empirical" methods are fully understood.

Minor issues:

Describe how the aridity index based on WorldCLIM data is defined.

Smith, T., Boers, N., & Traxl, D. (in review). Satellite-based evidence for global loss of vegetation resilience. -> either this study should be published first, or the empirical method and its accuracy should be described in full in the current paper.

Line 256: "we discard any perturbations where the time series does not drop significantly." How is "significantly defined?"

Interpretation of VOD as indicator of density and not of productivity. Previous studies (e.g. Moesinger et al., 2020; Teubner et al., 2019) have shown that VOD, especially in Ku-band, are actually closely related to vegetation productivity.

Figures 2 and 3 (and the corresponding plots in the supplement) have large interquartile ranges. The differences observed between bins are within these ranges. How statistically different are these differences?

It is unclear to me which IGBP classes were finally used. How much impurity did you allow? Were managed lands excluded? A map showing which pixels were finally used, and which not, would be very useful here.

Reviewer #1 (Remarks to the Author):

The submitted paper by Smith and Boers evaluates the 'resilience' of different land cover types globally. Vegetation resilience has many definitions throughout the literature, and here the authors use time series of VOD and NDVI to determine the rates at which ecosystems return a de-trended pre-disturbance mean state. The authors present four different ways to calculate the same metric, their 'recovery rate', and how this metric varies with aridity (improperly defined) and precipitation variability. A few things that should be addressed:

While the metric itself, and how it is calculated, are interesting and novel, much of its development as a new approach as well as the global patterns of this new metric are already in review in another paper that is included here. Many studies have examined vegetation resilience through other metrics, and a better case why this new metric is needed should be made. As given in the methods (which is missing some definitions for symbols used) the 'recovery rate' should be proportional to the natural log of the autocorrelation, which. More effort needs to be made justifying why another metric, that scales with ones presented in other studies, should be studied.

Thank you for these comments. We agree that several authors have used lag-one autocorrelation and variance as vegetation resilience metrics. However, their use was motivated purely by theoretical considerations stemming from how theoretical systems move towards or away from bifurcation-induced transitions. Our work – recently published in Nature Climate Change (<https://www.nature.com/articles/s41558-022-01352-2>) – shows for the first time that there is an empirical basis for using these metrics to measure vegetation resilience. However, it should also emphasized that the approximation of the actual recovery rates that is obtained by taking the natural logarithm of the AC1 is far from perfect (see Fig 2 in Smith et al., Nature CC 2022); it is therefore crucial to show the empirical recovery rates side by side with the theoretical metrics based on AC1 and Variance for comparison as we do in the present study. We chose to include several metrics here to be as transparent as possible and to show that the relationships that we find are not solely driven by which definition of resilience we use (e.g., based on internal variability, based on system memory, etc), but are concise across different ways of defining resilience, although subject to some small-scale differences. We have revised our introduction and methods to be more explicit about this justification and to make the distinction to previous studies clearer.

It seems to me that the main significance of this study is relating the authors' new metric to precipitation and aridity. For this, based on the current presentation, it is not clear how much of a control the climatic drivers exert relative to the vegetation itself. For example, does the precipitation have a similar 'recovery rate' as the vegetation? I.e., is what is being reported the recovery of the climate or of the vegetation. These may not be able to be disentangled, but the point needs some clarification.

This is indeed an important point to address – the response of vegetation to precipitation changes is clearly highly variable in space and time. The theoretical framework we use for

vegetation recovery does not really fit to precipitation – precipitation does not include the same level of system memory or variability (e.g., the previous vegetation state directly influences the future one, precipitation is not as tightly linked on monthly to annual time scales). It is not possible to calculate a precipitation ‘recovery rate’ within the same framework, also because there are no abrupt transitions following perturbations in precipitation, in contrast to vegetation. It would hence also be impossible to compare these rates in space and time.

Furthermore, it is exactly that signal of ‘change in environmental conditions’ that should be expressed in the resilience metrics (e.g., AC1 and variance), even absent an abrupt transition (which we only quantify with our empirical metric). Those areas which are less responsive to changes in climate (e.g., better buffered against changes, more resilient) are exactly what we quantify with our theoretical metrics. To truly answer the proposed question about the influence of climatic changes, we would have to use a time series approach for both variables, and attempt to link changes in vegetation resilience to explicit drivers. While this is an interesting research question (indeed, it is a topic we are working on!), it answers a different question than what we present here – a concise global relationship between water availability and *long-term* vegetation resilience. We have added additional clarification to this point in-text, and added a caveat that we do not look at any long-term changes in this manuscript (please also see the Discussion section).

As a reader of this study, some of the choices that the authors have made are unclear to me. Details as to why VOD or LAI are both used are not elaborated well. Why present both, does one present more, or different information than the other? The same can be said about the different precipitation metrics. Similarly, while the authors claim to have 4 different metrics of resilience, they all can be derived, or result in, the same conceptual model of exponential recovery of vegetated ecosystem post disturbance. In particular, for the 3 models derived from the assumption of an Ornstein-Uhlenbeck process, which of these three best captures this metric, are differences expected between the three? If something truly adheres to this type of process (which can be simulated easily) which of these three is expected to give the best estimate of the true parameter?

In terms of the two vegetation datasets, they certainly contain different information. VOD is more sensitive to vegetation density and water content, while NDVI is more sensitive to photosynthetic activity or chlorophyll content. This means that we do not expect identical recovery rates for the same system and the same disturbances. That being said, the *scaling* of resilience in the two vegetation datasets can be compared – both measure vegetation response to some outside pressure. We thus use these two independent, state-of-the-art satellite products mainly to establish robustness of our results, and indeed, the global-scale resilience patterns we discover (e.g., Figs. 2-4) broadly agree across the two vegetation datasets.

Similarly, each of the four resilience metrics that we compute measure resilience with slightly different assumptions. The empirical recovery rate is directly estimated from measurements and should be therefore seen as a baseline; it is however, by construction, not available for locations that did not experience large-scale shifts from which to recover from. The other three,

theoretical, metrics can be derived based on the assumption that the vegetation can be locally linearized. Note that these equations do not require that the entire vegetation time series can be approximated as realization of an Ornstein-Uhlenbeck process, but only that the systems under study are close to equilibrium and can hence be locally linearized around this equilibrium. We included all four for robustness – if several different ways of calculating resilience agree, we are more confident in our overall results. Since the regression-based estimate of the recovery rate can be analytically shown to be very close to the AC1-based estimate (which we confirm empirically), we have decided to omit this fourth metric in order to focus on truly independent ways of estimating the recovery rate. We have expanded upon our reasoning for including all three remaining metrics (the empirical one as well as the Variance- and the AC1-based one) in the Introduction.

Finally, while of minor import to the conclusions in this paper, the Aridity Index provided by WorldCLIM (Trabucco & Zomer 2019) is the inverse from what is commonly understood in hydrology and climate science, where this quantity is typically calculated as PET/P not P/PET. Aridity is defined as PET/P because it is a measure of how arid a location is, with increasing values corresponding to more arid locations. P/PET is a wetness index. Aridity defined as PET/P is consistent with Bydco's framework delineating water and energy limits for terrestrial systems, though how to specify PET is debated. See Greve et. al. and many of the papers within (DOI: 10.1088/1748-9326/ab5046) for more details. Please clarify this point in the present study.

Thank you for this observation! We have updated our text and figures to conform to the more commonly-used aridity definition, and have also clarified this in the figure descriptions.

A few other minor considerations:

- Consider expressing the recovery rate as a time value instead of a rate. The inverse of the recovery rate could easily be converted into recovery rate half-life and presented in units of months or years. This might help readers more easily understand the presented results.

The referee is of course correct that the inverse of the recovery rate yields a characteristic time scale of recovery. However, we would like to argue that the recovery rate is the more direct measure of resilience, and more generally a direct measure of stability in dynamical systems. Essentially, thinking of the dynamics as driven by a potential gradient, the recovery rate would then directly correspond to the steepness of the potential well, which directly translates to the strength at which the system is restored to equilibrium.

- Be very clear that you're using the inverted definition of aridity. Explicitly state P/PET in the figure legend

This has been updated.

- More details on the sample sizes, resolutions in space and time, about the data products and the confidence in the derived metrics need to be given.

We agree! Spatial and temporal resolutions of each data set are listed in the 'Data' section of the Methods. We have moved the description of the precipitation data upwards to make this clearer. Further details about the statistical reliability of our derived relationships are listed in the Results section, with a description of the novel statistics found in the Methods section.

- Rainforests are not water limited but energy limited, so more rainfall leads to less productive systems, this might explain some of your anomalous results for very wet systems

Thank you, this is a very good point! We have added this possibility to the discussion section of our manuscript.

- No consideration for the interaction between precipitation and aridity is presented and might be worth considering. These are not independent variables. Wet and dry systems might react differently at different levels of precipitation variability.

This point is well taken – we struggled initially to disentangle these two signals. While we feel that our normalized precipitation metric covers some of this difference, we want to further split these signals. In the updated MS, we have included a new figure which attempts to disentangle these results, and we also discuss this in more detail in our revised manuscript. In particular, see Supplemental Figures 4-5, which present the same data but for a fixed precipitation window (by precipitation percentile, for each individual land cover type). Our inferred relationship between resilience and precipitation variability is robust for two different ways of controlling for precipitation volume (simple division by mean annual precipitation and fixing our analysis over a limited precipitation range).

- If there is not clear reason to choose one of these metrics over another, then would a multi-metric average be more reliable?

The different metrics do not necessarily sit on the same 'scale' and are not cross-comparable. 'Resilience' in this case is not a fixed-magnitude number, but rather a relative metric – a multi-metric average does not have support within the mathematical framework we employ.

Overall, I think the study and its approach are interesting and do not see any major flaws in the methodology. More details are needed to help make the case for the importance of this study, how it separates itself from prior work, how the choices of metrics and their calculation were decided, and how the observed patterns relate to overarching temporal patterns in climate.

- Stephen Good

Thank you for these detailed and helpful comments!

Reviewer #2 (Remarks to the Author):

The authors employ both theoretical and empirical resilience indices to quantify how precipitation availability and variability impact vegetation resilience at a global scale. This is an interesting study that shows an increasing resilience of vegetation with aridity across different ecosystems. By contrast, the grasslands showed decreased vegetation resilience under high interannual precipitation variability. This finding is important because the increasing precipitation variability will lead to critical challenges for the state and stability of vegetation systems. The manuscript is clearly written, and the key message is nicely illustrated. However, the evidence (or results) to support the conclusions is simple and weak, and it is too subjective to interpret the results in this manuscript. I have the following comments for the authors to consider:

Thank you for taking the time to review the MS. We are glad that you see the importance of the conclusions we draw, and have endeavored to clarify and strengthen our support for them in this revision.

1. *The authors quantify resilience as a function of aridity and precipitation interannual variability. How about the relationships between interannual variability parameters and aridity? Whether the contrasting effects of precipitation availability and variability results from the negative correlations between precipitation availability and variability? For instance, in the grasslands, the annual precipitation is lower and the relative deviation from precipitation is generally higher. In comparison, the precipitation is high and evenly distributed over the years in the tropics. In other words, whether they do necessarily show a contrasting behavior?*

Thank you for this very important point. While dry regions often have higher inter-annual precipitation variability, there is not always a clear relationship between ecosystem and precipitation variability. For example, in contrast to intra-annual variability, relative **inter-annual** precipitation variability is similar in the Amazon, Siberia, and the Pacific Northwest (Figure 1 of this Reply).

Figure 1 - Global Relative Inter-annual precipitation variability (1981-2021). Calculated as the mean annual precipitation standard deviation divided by mean annual precipitation.

To try to disentangle these effects further, we have split precipitation variability into two factors – one intra-annual and one inter-annual (cf. Figures 3,4). Furthermore, we have also performed

our analysis using a fixed mean annual precipitation range (based on percentiles of precipitation in each individual land cover type, Supplemental Figures 4-5). Both means of limiting the influence of absolute precipitation amount yield similar results.

2. The title, the abstract, and the study objective all clearly state that the interest here is how key dimensions of precipitation regimes affect vegetation resilience. My concern is that in the manuscript, effects in precipitation amount and interannual variability dominate the results, but precipitation variability includes many other components as well.

In our updated manuscript we have endeavored to include additional discussion of precipitation distributions, both inter- and intra-annual. To quantify seasonality, we use the Walsh-Lawler seasonality index, which quantifies how well-distributed precipitation is throughout the year. For inter-annual variability, we use the standard deviation of annual precipitation sum, normalized by the long-term average precipitation sum to control for differences in absolute precipitation across regions. Both of these two metrics yield similar results – a general decrease in resilience with more variable precipitation. We have added additional discussion of this difference and its implications in the updated manuscript.

Furthermore, we have removed the discussion of precipitation frequency in favor of our normalized inter-annual precipitation metric for simplicity. Both precipitation standard deviation and precipitation frequency are highly correlated to mean annual precipitation itself (Figure 2 of this reply). However, precipitation standard deviation is straightforward to normalize with precipitation sums, as opposed to precipitation frequency which does not share a unit with mean annual precipitation. We feel that the combination of a seasonality index for intra-annual and the normalized variability for inter-annual variability captures the key elements of precipitation that we wish to explore with regards to resilience.

Figure 2 - Mean annual precipitation compared to standard deviation of annual precipitation and average precipitation frequency. Both metrics show a strong linear relationship, indicating that mean annual precipitation is a strong control on non-normalized precipitation variability measures.

3. The authors believe the similar pattern shown in Figure S3 & S6 and Figure 2 stem from the spatial coherence between mean annual precipitation and the three components of precipitation

variability (i.e., the standard deviation of annual precipitation, annual precipitation frequency, and seasonality). All existing precipitation metrics are correlated to some extent. However, these metrics represent different mechanisms (drought, seasonality, and precipitation irregularity). In my opinion, interannual variability alone does not retain the full spectrum of information on precipitation variability. Why not use relative variability estimates to describe other aspects of precipitation variability? How vegetation resilience is affected by intra-annual variance of precipitation?

In our revision, we have replaced our measure of seasonality based on the wettest and driest month with the more commonly used Walsh-Lawler seasonality index which better captures the distribution of precipitation throughout the year. As you suspected, intra-annual precipitation variability shows some of the same impacts on vegetation resilience as inter-annual precipitation variability. This analysis has been included in the updated manuscript.

4. Figure 2 & 3: a robust statistic is required to quantify the response of vegetation resilience to precipitation availability and variability instead of describing the resilience trend only. The authors should stress out the statistical robustness of this study in the method section. For example, the error bars shown in Figure 2 are very long, so I am curious that whether the correlations are significant for most land cover types. In addition, all correlations in this manuscript only presented crude trends (e.g., decrease or increase), there are no fitted equations, R²-values, and even P-values were provided. Therefore, it is unconvincing, in terms of me, on assessing the correlations between vegetation resilience and precipitation availability and variability.

We fully agree with the referee! In our revision, we have computed non-parametric trend values (Kendall's Tau), and added their statistical significance in terms of p-values. As you expected, not all land cover types maintain a significant relationship – however, the majority do. We have added the results to the legends of Figs. 2-4, updated our Methods to include this testing, and modified our discussion to include only results we are highly confident in based on the performed statistical tests. As a final check, we also use a Monte Carlo approach to randomly sample each bin and check the direction of the computed Kendall-Tau statistics. We report the fraction of randomly selected binned lines which share a sign with the Kendall-Tau of the median line as well.

5. All the results seem to rely on the binning analysis but the methods for this binning issue are not included in the study Methods section. The precipitation bin used in this study might bias the trend of vegetation resilience. I would like to see sensitivity tests to the bin size. Whether bin size change the trend of raw data? or whether each bin had enough samples?

This is a very good point – all of our results are indeed potentially sensitive to that binning. We have thus re-run our analysis with varying bin sizes, and included this robustness analysis in the Supplement. We find similar results using different bin slices (along Aridity: 0.025, 0.05, 0.1), albeit with different Kendall's Tau trends. Generally, smaller bin sizes yield more statistically significant Kendall's Tau statistics, as expected from the high number of resulting data points to

compute Kendall's Tau from (which comes at the cost of fewer samples within each bin, of course). We limit our analysis to bins with at least 50 members (to compute a reliable median) and only report Kendall-Tau statistics for median lines with at least 10 points (to reduce errors from poorly sampled data). We have included further discussion of our binning in the Methods section.

6. The descriptions on correlations between vegetation resilience and precipitation availability and variability are so simple, e.g. only decrease or increase. The correlation between restoring rate λ and aridity (Figure 2), for example, presents remarkable non-linear regression, e.g., keeping relative stable after about 0.6-0.75 (visual estimate), which is near the threshold that distinguishes the Semi-arid and Dry sub-humid (or Humid) (Trabucco and Robert 2018, UNEP 1997). If we considered the threshold, the results might correspond to another different story.

Thank you for this observation! In our revision, we have added additional discussion of non-linearity in our discovered relationships. Note that these nonlinearities have also led us to use Kendall's Tau instead of linear trends to measure the dependencies.

7. It is unbelievable that the authors explain the inconsistent patterns in evergreen broadleaf forests by "data unreliable" in lines 82-83. If the data of evergreen broadleaf forests are unreliable, how to prove that the data for other land cover types are reliable. Therefore, it is inadvisable to deal with unexpected results in this way. This interpretation gives me a sense that it doesn't matter what the results are, we would or should only keep the conclusions are consistent with common sense.

It is well-known that NDVI saturates in very dense vegetation settings. This does not imply that NDVI is useless globally, but rather that it needs to be treated with care over dense vegetation settings. This is part of the reason that we use two vegetation datasets – VOD does not saturate over dense forests to the same degree, though it also eventually loses product skill over very dense or very wet areas. Much of the broadleaf evergreen (orange line, Figure 2) region falls exactly in this region where NDVI is known to saturate. Furthermore, there are minimal regions of precipitation deficit over this land cover, meaning that a significant portion of this land cover type is energy rather than water limited. This will of course modify our inferred relationship as well.

We have expanded our discussion of possible reasons for this result in our revised manuscript.

8. Line 154-161: The aspect of plasticity is not discussed in the paper. There are many mechanisms for plants to adjust to changing environmental conditions and lower vegetation resilience of grass-dominated regions may also reflect a structural adjustment to higher interannual precipitation variability.

This point is well taken – we do not go into detail on the various potential mechanisms for plants to respond to changing climate conditions. However, it should be clarified that directly comparing the resilience between different land-cover types is difficult – the characteristic

recovery time of say, a dense rainforest and a savanna are very different. Hence, some of the differences between biomes/land covers is simply a reflection of different intrinsic timescales of recovery for those different types of vegetation. The point we want to make is that *within a single land cover class*, where vegetation mixes/recovery times should be more equal, we still see a loss of resilience with changing water availability.

Within grasslands, for example, there likely remain structural adjustments that could modify that characteristic recovery time scale (e.g., different plant mixes in different rainfall regimes, or changes in plant structure/phenotype). Disentangling which adjustments are positive or nominal in terms of resilience (e.g., equal recovery after shock) and negative (e.g. do not help or even hinder recovery) would require much more fine-scale analysis. In this revision, we have added additional discussion of caveats related to analyzing by land cover, and possible plant responses to water availability.

Other comments:

1. The authors select both NDVI and VOD data to quantify vegetation resilience patterns. Why the spatial distribution of VOD is presented in Figure 1 but keep the NDVI in Figure S1?

We chose to only show one of the two grids for simplicity's sake – the global spatial patterns are similar. Following the referee's advice, we have added an NDVI map back into the main MS in this revision.

2. The source of precipitation data should also be included in the Methods.

The data source was mentioned (line 231), but not in the 'Data' section. We have made the data source more apparent earlier in the Methods section.

3. Figure 2, 3 and everywhere: Please give the results of linear regressions between aridity and λ . The error bars shown in Figure 2 are very long, so I am curious that whether the correlations are significant for most land cover types?

We have chosen to report Kendall-Tau statistics here, as the relationships are clearly non-linear. These have been added to each figure and discussed throughout.

4. Line 146: Supplemental Figure S3?

This has been fixed.

Reviewer #3 (Remarks to the Author):

This study looks at the resilience of global ecosystems as a function of aridity and precipitation (variability). The topic is of high relevance and the methodology (apart from the empirical method submitted elsewhere by the lead author) well established. There is novelty in the fact that apart from assessing resilience worldwide and using times series of AVHRR NDVI, the study uses a new, long-term vegetation optical depth dataset from passive microwave observations.

Even though the study is performed in a sound way and very well written, I have strong concerns about the conclusions drawn from it. I am even more worried about the conclusions drawn by the study submitted by the authors elsewhere, i.e.: "Smith, T., Boers, N., & Traxl, D. (in review). Satellite-based evidence for global loss of vegetation resilience.", which unfortunately is not publicly accessible.

The issue is that looking at such delicate statistical metrics from long-term observations assumes that these datasets have homogeneous characteristics in space and time. As correctly noticed by the authors, NDVI is known to saturate for denser canopies. But even for microwave observations, which have much longer wavelengths, this may be the case. Typically, the signal-to-noise ratios of remotely-sensed vegetation datasets are optimal at intermediate vegetation densities, when confounding background (soil) effects are relatively low but the observations do not yet saturate. These are vegetation types where the strongest response was found. Also in the temporal domain, retrieval skills are not static with newer satellites typically being less noisy. But, more importantly, merged datasets like the VODCA product used in this study, have a strongly variable number of observations used per satellite-blending period, which has a direct impact on the noise, and hence metrics like AR1.

As mentioned above, I am particularly worried that the "loss-of-vegetation-resilience signal" seen in the paper that is in review is simply a result of the increased product skill over time.

In general, the quality of the current paper can only be fully assessed once details on the "empirical" methods are fully understood.

Thank you for taking the time to review our MS, and apologies that you did not find our previous manuscript which was among the included review materials. That paper has now been published (<https://www.nature.com/articles/s41558-022-01352-2>) and includes discussions of possible caveats associated with using merged products like VODCA.

With regards to your point about different signal-to-noise ratios for different vegetation types, we fully agree – hence we assess each land-cover type individually in the present paper. All else being equal, the signal-to-noise ratios of similar biomes should be comparable, and hence useful for finding other factors (e.g., precipitation) that influence resilience. We have added a discussion of this to our caveats section.

We are aware of the merging of different sensor data to obtain the VODCA dataset, and how this may lead to nonstationarities of the autocorrelation and variance of the resulting time series. This is the subject of an in-progress collaboration with the authors of the VODCA dataset, where we explore to what degree other data pre-processing steps – such as detrending and deseasoning the time series – influence the final correlation structure and hence AC1 and variance. In short, in our previous paper, we find significant trends exactly during times of constant satellite mixes when, all else being equal, changes in input data and blending procedures should not be responsible for shifts in variance and autocorrelation.

It should be emphasized, however, that concerns regarding nonstationarities due to the merging of different satellite sensors only apply when considering changes of AC1- or Variance-based resilience estimates over time. In the manuscript under consideration here, this is not the case; moreover, the empirical confirmation that the AC1 and Variance can be used to estimate resilience, as obtained in Smith et al., Nature Climate Change 2022, is also independent of possible non-stationarities of higher-order statistical moments in the VODCA time series.

Please note that the details of the empirical method – and our assessment of data limitations in terms of long-term trends – are described in much more detail in our previous manuscript, which can be found here: <https://www.nature.com/articles/s41558-022-01352-2>

Minor issues:

Describe how the aridity index based on WorldCLIM data is defined.

This has been updated after your and other reviewer comments.

Smith, T., Boers, N., & Traxl, D. (in review). Satellite-based evidence for global loss of vegetation resilience. -> either this study should be published first, or the empirical method and its accuracy should be described in full in the current paper.

Thank you, this paper has since been accepted as mentioned above. The final title of the study is “Empirical evidence for recent global shifts in vegetation resilience”

Line 256: "we discard any perturbations where the time series does not drop significantly." How is "significantly defined?"

Details on our empirical method can be found in the other paper which has since been accepted. In short, we look for drops of at least 0.01 – this is not a variable threshold, but was found to significantly reduce poorly-defined fits during our testing of our algorithm. It should be noted that this threshold is not used in isolation, but as part of several metrics (Kolmogorov-Smirnov tests, fit statistics, percentile thresholds). We have modified our Methods text to more concisely describe our empirical metric, with reference to our previous work.

Interpretation of VOD as indicator of density and not of productivity. Previous studies (e.g. Moesinger et al., 2020; Teubner et al., 2019) have shown that VOD, especially in Ku-band, are actually closely related to vegetation productivity.

Thank you for this point. Indeed, VOD is of course tied to vegetation productivity; we have made this clearer in the revised manuscript. The point we wanted to make here is that NDVI and VOD do not respond at the same speed to vegetation changes – particularly after a shock that removes significant vegetation (e.g., regrowth after a fire). These differences – alongside simple differences in the noise structure of the two data sets driven by temporal sampling, the wider range of values for VOD, etc – will naturally cause some differences in the computed AC1 and variance. However, our results are broadly consistent for the two different datasets, which in our opinion indicates that the inferred relationships between resilience on the one hand, and aridity and precipitation variability on the other, are robust.

Figures 2 and 3 (and the corresponding plots in the supplement) have large interquartile ranges. The differences observed between bins are within these ranges. How statistically different are these differences?

In our revision, we have computed Kendall-Tau statistics with a Monte-Carlo approach. In short, we retrieve 1000 random samples – distributed over the given bins – and compute Kendall-Tau values. We then compare the Kendall-Tau statistic computed over the binned medians to the 1000 random samples to determine the reliability of our relationships. In short, we report the number of Kendall-Tau values from our random samples that have the same *direction* (e.g., pos/neg) as our Kendall-Tau value for the median. We have updated our Figures and Methods to reflect this additional check.

It is unclear to me which IGBP classes were finally used. How much impurity did you allow? Were managed lands excluded? A map showing which pixels were finally used, and which not, would be very useful here.

The classes listed on the legends of the figures were used – so, IGBP 1-10. We also remove significant human-influence land cover, e.g., Line 210. We also remove land covers with less than 1000 pixels from our analysis (Figures 2-4). We have added a map of the land cover classes (color-coded to correspond to the lines in Figures 2-4 and the Supplement) as a panel on Figure 1.

Reviewer comments, second round -

Reviewer #2 (Remarks to the Author):

This is a revised version of an earlier rejected manuscript, which found that the roles of precipitation availability and variability are contrasting for global vegetation resilience. The authors have made substantial revisions to fix the critical comments from three reviewers. After reading the revised manuscript and their replies to all comments, I found the manuscript has been improved. However, some new questions emerged during the reading. Please find them as follows.

1. The revised version changed the key factor from precipitation to water availability and variability. It is well known that precipitation does not equal to available water for plants. Some recent studies use soil moisture or water storage instead of precipitation to explore the role of water availability in vegetation dynamics.
2. The quantitative method of resilience matrices, especially the theoretical method, has been recently published on NCC (<https://www.nature.com/articles/s41558-022-01352-2>). In the NCC-2022 paper, the authors have reported that the temporal change in vegetation resilience is non-linear from 1992 to 2017. There is a significant shift towards a global decline in vegetation resilience since the early 2000s. It must be very important to present new advances in this revised version, especially based on the NCC-2022 paper.
3. If the key finding of this study is that global vegetation resilience links to water availability and variability, an important question could be how water availability and variability jointly determine the vegetation resilience on the global scale. Some analyses on the relative roles of those two water indices could be helpful.
4. There is large difference in the relationship between λ and water indices between NDVI and VOD. The authors mentioned that those analyses showed consistent patterns across land cover types. Some additional analyses to show their difference are also very important.
5. This study applied all the analyses based on the IGBP land cover classification. One large uncertainty is the human activity, e.g., cropland expansion, which could affect some conclusion. I know it's difficult to avoid such uncertainty in this study, but some analyses on the cropland and land management (e.g., irrigation) are helpful to address such uncertainties.

Reviewer #3 (Remarks to the Author):

The authors have put a lot of effort into improving the manuscript. Besides, since the first submission of the manuscript, two other papers on a similar topic (studying resilience from Vegetation Optical Depth and other Earth observation data) with involvement of the same authors have been published in Nature Climate Change. This helps the interpretation of the results of the current paper. The methodology and conclusions of the two other papers recently published in Nature Climate Change are heavily disputed, as they implicitly assume that the datasets used have stationary higher order characteristics, which is not true. Although similar metrics and data are used for the current NCOMMS paper, the heterogeneity of the dataset is not as critical here because it does not look at changes in resilience. However, it needs to be looked at how the heterogeneity in space and time impacts the patterns found since also the "sharp transitions in the vegetation time series" may be impacted by this.

I appreciate that resilience was assessed against the Walsh-Lawler seasonality index, however the relationships shown in the plots (Fig. 4) are not as convincing as described by the authors. The suggested increase in resilience with a more even distribution of precipitation throughout the year is only observed in few classes. This needs to be discussed in a more objective way.

Reply to Reviewers – Second Round, Nature Communications

Reviewer #2 (Remarks to the Author):

This is a revised version of an earlier rejected manuscript, which found that the roles of precipitation availability and variability are contrasting for global vegetation resilience. The authors have made substantial revisions to fix the critical comments from three reviewers. After reading the revised manuscript and their replies to all comments, I found the manuscript has been improved. However, some new questions emerged during the reading. Please find them as follows.

Thank you for the overall positive assessment of our revised manuscript; your new questions helped us significantly in further improving our manuscript.

1. The revised version changed the key factor from precipitation to water availability and variability. It is well known that precipitation does not equal to available water for plants. Some recent studies use soil moisture or water storage instead of precipitation to explore the role of water availability in vegetation dynamics.

This is a good point – getting directly at plant-available water is, however, difficult at the global scale. We feel that aridity is a much better proxy for water availability over the longer-term than mean annual precipitation, which has been used in previous studies. Getting at shorter-term water availability is somewhat more difficult – soil moisture sensors and models have a much higher degree of uncertainty than precipitation data. However, for completeness, we have re-calculated our two precipitation variability metrics with ERA5 soil moisture data. We use the sum of soil moisture from the surface down to 28cm of depth (first two layers of the ECMWF Integrated Forecasting System soil moisture estimates). These additional analyses are in very good agreement with the results we obtain for aridity and therefore provide a data-wise independent corroboration of our results; we have therefore added this as additional SI figures and also mention this in the revised main text.

For reference, we include in this Reply the relationship between mean annual soil moisture and resilience, calculated by land-cover class as Figure 1.

Figure 1 – Mean annual soil moisture (1981-2020) compared to vegetation resilience. Lower soil moisture is associated with less resilient vegetation.

2. The quantitative method of resilience matrices, especially the theoretical method, has been recently published on NCC (<https://www.nature.com/articles/s41558-022-01352-2>). In the NCC-2022 paper, the authors have reported that the temporal change in vegetation resilience is non-linear from 1992 to 2017. There is a significant shift towards a global decline in vegetation resilience since the early 2000s. It must be very important to present new advances in this revised version, especially based on the NCC-2022 paper.

We feel that these two publications address distinct questions, each of significance to our understanding of global vegetation dynamics. In the NCC-2022 paper, we focused on long-term changes, and did not explore any drivers of those changes, or potential controls on resilience changes due to climatic factors. In this new work, we instead focus on that key question – how do global scale vegetation resilience patterns respond to water availability and precipitation variability? Here, we do not look at any long-term changes, but rather describe and analyze patterns both at the global scale and broken down by land cover type, to improve our understanding of the dependence between vegetation resilience on the one hand, and water availability and variability on the other hand. We thus feel that this new work is complementary to our previous work and advances it toward process-based explanations of resilience changes. Our methodology in the present manuscript builds upon the empirical resilience estimation methodologies we introduced in our NCC-2022 paper, but the

similarity between the results presented in this manuscript for the empirical and the theory-based recovery rates also adds further independent evidence that the AC1 and variance can indeed serve as measures of vegetation resilience.

3. If the key finding of this study is that global vegetation resilience links to water availability and variability, an important question could be how water availability and variability jointly determine the vegetation resilience on the global scale. Some analyses on the relative roles of those two water indices could be helpful.

Thank you for this comment. As per your suggestion, we looked more closely at the direct relationship between resilience and the joint effects of water availability and variability (both intra- and inter-annual). We found that aridity (long-term water availability) was the strongest control on resilience, followed by inter-annual precipitation variability. Intra-annual variability presents more of a mixed picture – some regions with highly seasonal precipitation nevertheless are highly resilient. In particular, throughout the Sahel and other dry grass/shrublands, plants are well-adapted to short precipitation periods, and thus maintain high levels of resilience even in quite arid environments.

These relationships have been further explored and displayed in an additional Figure (Figure 2 of the revised manuscript), also included below.

Figure 2 - Relative importance of intra- and inter-annual precipitation variability compared to aridity. (A,B) Vegetation optical depth (VOD), (C,D) GIMMS3g normalized difference vegetation index (NDVI), and (E,F) MODIS NDVI. Aridity compared to intra-annual (left column) and inter-annual (right column) precipitation variability. Hexbins colored by recovery rate computed from AC1 (minimum five points per bin). Transition from water surplus (aridity < 1) to deficit marked with dashed vertical line; there is a sharp decrease in recovery rates as water availability increases. Higher inter-annual precipitation variability (right column) consistently leads to less resilience, intra-annual precipitation variability, i.e. seasonality, has a more varied impact.

4. There is large difference in the relationship between λ and water indices between NDVI and VOD. The authors mentioned that those analyses showed consistent patterns across land cover types. Some additional analyses to show their difference are also very important.

In response to this and comments from the other reviewer, we have added a second NDVI data set from MODIS to our analysis. All three data sets show similar relationships, with increasing water availability across time scales driving increased resilience. While it is clear that the scaling of resilience metrics between data sets is different, we attribute this primarily to differences in the data sets themselves. Each data set uses different spatial, temporal, and spectral resolutions to develop their vegetation indices; these differences naturally lead to varying resilience estimates. Furthermore, NDVI (a measure of chlorophyll content) and VOD (a measure of vegetation density) do not measure the same variable, but rather complementary metrics derived from the same vegetation. The fact that all three data sets show the same broad-scale patterns makes us confident in our results. In particular, the newly added MODIS NDVI data assure that the results we reveal here are not influenced by changing measurement sensors over time, which is a caveat of our NCC-2022 study.

5. This study applied all the analyses based on the IGBP land cover classification. One large uncertainty is the human activity, e.g., cropland expansion, which could affect some conclusion. I know it's difficult to avoid such uncertainty in this study, but some analyses on the cropland and land management (e.g., irrigation) are helpful to address such uncertainties.

Thank you! We have updated our description of our Methods to make it clear that we remove any land-cover that has changed during the last 20 years, to try to as conservatively as possible remove human-influenced land cover. For example, if a forest changes to grassland and back to forest, we do not use that pixel in our analysis. This also helps remove any pixels that MODIS has trouble classifying (e.g., grassland vs agriculture), as these pixels tend to change classification year-to-year.

Reviewer #3 (Remarks to the Author):

The authors have put a lot of effort into improving the manuscript. Besides, since the first submission of the manuscript, two other papers on a similar topic (studying resilience from Vegetation Optical Depth and other Earth observation data) with involvement of the same authors have been published in Nature Climate Change. This helps the interpretation of the results of the current paper. The methodology and conclusions of the two other papers recently published in Nature Climate Change are heavily disputed, as they implicitly assume that the datasets used have stationary higher order characteristics, which is not true. Although similar metrics and data are used for the current NCOMMS paper, the heterogeneity of the dataset is not as critical here because it does not look at changes in resilience. However, it needs to be looked at how the heterogeneity in space and time impacts the patterns found since also the "sharp transitions in the vegetation time series" may be impacted by this.

Thank you for this comment – it is indeed important to consider changes in the higher-order statistics of the individual data sets when assessing resilience. As you noted, this is much more important for

assessments of changes in resilience than for our current analysis which is based primarily on long-term calculations. Please also see our recently submitted paper investigating the effects of changing sensor mixes through time in detail here:

<https://esd.copernicus.org/preprints/esd-2022-41/>

To further address this concern in the context of the present manuscript, we have added a third data set to our analysis, namely MODIS MOD13 NDVI. This data is drawn from a single instrument record, and thus is not influenced by the same changes in statistics as VOD and AVHRR. It has the additional benefit of being a higher spatial resolution, which gives us more data to examine.

While the data do not match perfectly, all three data sets show the same broad scale patterns – increasing water availability leads to improved resilience, increasing inter-annual precipitation variability leads to reduced resilience. We thus feel that our analysis remains robust, and the addition of a single-instrument record improves the reliability of our results.

I appreciate that resilience was assessed against the Walsh-Lawler seasonality index, however the relationships shown in the plots (Fig. 4) are not as convincing as described by the authors. The suggested increase in resilience with a more even distribution of precipitation throughout the year is only observed in few classes. This needs to be discussed in a more objective way.

Thank you for this comment – indeed, we slightly over-interpreted those results. While most land cover types maintain a positive Kendall-Tau statistic, indicating decreasing resilience with more seasonal precipitation, this relationship is not as strong as that for aridity or inter-annual precipitation variability. We have modified our discussion of this figure, and re-focused our analysis on what we feel that we can strongly show: water deficits over multi-year time scales lower vegetation resilience.

Reviewer comments, third round -

Reviewer #2 (Remarks to the Author):

I appreciate that the authors carefully revised their manuscript based on my comments. I had two major concerns about the earlier version of the manuscript: (1) the difference between this study and their NCC-2022 paper, and (2) the relative role of water availability and variability. Overall, I think the authors have solved my major concerns:

For the first concern, the authors have clearly explained it in their response letter. The current study's unique contribution is how water availability and variability differently affect global-scale vegetation resilience patterns. After reading the new version, I found the authors only cited their NCC-2022 paper to highlight the method rather than their major findings in that study. A short introduction to the long-term trends of global vegetation resilience is important to guide the readers to understand this study better.

For my second concern, the authors have added new analyses (e.g., Figure 2) to quantify the relative importance of intra- and inter-annual precipitation variability. This analysis adds some novel findings to this study. For example, it is interesting that the dependence of intra-annual variability upon aridity (Panels a and c) is much more divergent than that of inter-annual variability (Panels b and d). I'm not sure if the divergent occurrence of extreme drought events during the last decades (e.g., Fig S3 in Huang & Xia 2019, GCB) could be a reason. I'd like to suggest that the authors discuss the role of regime shifts in extreme hydrological events in the "Intra- and Inter-annual Precipitation Variability" section.

Below are some new specific comments:

1. This study focuses on the variability of vegetation resilience, so the vegetation data must be detrended before the analyses. The authors have mentioned it, but how they detrended their data is unclear. Some sentences can be added in the method section to describe it.
2. The illustration of Figs. 3-5 is not very clear. One reason could be the overlaps between bars. The figure legends can be shown with less information. Also, the bars were not defined in the figure captions.
3. I think the method in this study could be useful for the community, but I didn't find a statement of the code availability in the current version.

Reviewer #4 (Remarks to the Author):

Summary:

This is a review of the manuscript Global Vegetation Resilience Linked to Water Availability and Variability (NCOMMS-21-44074B-Z) submitted to Nature Communications. The topic of this manuscript is well suited to the Nature Communications as it is an investigation of the climatic drivers of the resilience of ecosystems, in particular the rate of return to 'normal' after a large disturbance. This review is a review of an already revised paper and particular care was taken in considering the previous Reviewer 3's comments.

In the paper, the authors calculate resilience in three ways, as a coefficient that best fits the recovery from a large disturbance in timeseries of NDVI (2 products) and VOD. Metrics of water availability (primarily aridity) and precipitation variability (seasonal and interannual) are then used to explain global variation of vegetation resilience for all vegetation and binned by land cover type. The paper concludes that aridity is a large driver of resilience, while precipitation variability is an important but weaker driver. The paper is an interesting study of the connection of climate to resilience and the results discussed appear to be plausible. However, overall, I found the statistical treatment of the relationships between climate and resilience to be lacking or difficult to interpret. This is partly due to the difficult to read inclusion of the statistical results in the legends of figures. In addition, the concern that a change in statistics due to timeseries being made of multiple

instruments could be more strongly addressed. Detailed comments below.

Reviews of [figures] and [lines] below.

Major revisions:

[397 – 401] While the additional MODIS NDVI time series does support the robustness of the conclusions to a change in dataset, a more complete treatment of this potential issue is needed or the relevant results of the linked to study need to be detailed in the current paper and response. The link to the pre-review paper is not sufficient. If the resilience calculation were detecting either the change in instrument, or were finding significantly different resilience in one instrument time series vs. another it should appear in the time series of when the resilience events occur. One possibility is to show a distribution of when the resilience events being fit occur. Additionally, it would be useful to know how many points or events are happening in a particular location and how many events are being screened out based on the R2 metric [397 – 401].

[336 – 338][461-463][Figures 3 – 5, Supplemental Figures S4 – S11] This is a comment broadly on how statistics are used and displayed to support statements around 'robust trends' (here I am assuming this is the relationship between predictors and resilience, rather than through time) and 'break points'. The display items that support the relationship between climate variables and resilience contain so much information it is very difficult to interpret. While there do appear to be relationships there, the display figures, and paper as a whole, do not concisely show the statistical uncertainty of those relationships. In addition, the statistical results contained in the legend are nearly unreadable and very difficult to cross reference. I suggest summarizing these statistics in a few tables and potentially focusing on one dataset or one resilience metric in the main text that best supports the discussion while moving the rest to the supplement. To better support the discussion around a break point (aridity < 1) or the relative strength of relationships between resilience and the variability of resilience it needs to be clear how much of the data is explained by a linear model (trend) or potentially a parabolic or piecewise model (break point). I found it very difficult to understand which landcovers had significant relationships of resilience vs. aridity etc. and whether those relationships were statistically significant.

Reviewer #2

I appreciate that the authors carefully revised their manuscript based on my comments. I had two major concerns about the earlier version of the manuscript: (1) the difference between this study and their NCC-2022 paper, and (2) the relative role of water availability and variability. Overall, I think the authors have solved my major concerns:

Thank you for taking the time to review our MS once again. We are glad that we have addressed your major concerns! We will detail our response to your further questions below.

For the first concern, the authors have clearly explained it in their response letter. The current study's unique contribution is how water availability and variability differently affect global-scale vegetation resilience patterns. After reading the new version, I found the authors only cited their NCC-2022 paper to highlight the method rather than their major findings in that study. A short introduction to the long-term trends of global vegetation resilience is important to guide the readers to understand this study better.

We have added a reference to the findings in our earlier paper to our introduction to add context to this manuscript.

For my second concern, the authors have added new analyses (e.g., Figure 2) to quantify the relative importance of intra- and inter-annual precipitation variability. This analysis adds some novel findings to this study. For example, it is interesting that the dependence of intra-annual variability upon aridity (Panels a and c) is much more divergent than that of inter-annual variability (Panels b and d). I'm not sure if the divergent occurrence of extreme drought events during the last decades (e.g., Fig S3 in Huang & Xia 2019, GCB) could be a reason. I'd like to suggest that the authors discuss the role of regime shifts in extreme hydrological events in the "Intra- and Inter-annual Precipitation Variability" section.

Thank you for this comment. Overall, our findings suggest that year-to-year variability is a more important driver of vegetation resilience than seasonal variability. The divergent scaling in Figure 2 likely indicates differences in plant adaptations to water deficits at different time scales. Arid regions which receive small amounts of precipitation throughout the year will behave differently than those that receive the same total precipitation, but rather concentrated into a short time period. This is something that plants in different environments can adapt to over time, and still implies a quasi-annual or regular cycle to vegetation growth.

On the other hand, highly variable inter-annual precipitation implies multiple years with too little or too much precipitation. This is much harder to adapt to, especially over relatively short time scales (for example, as you mentioned with changing drought dynamics). We posit that this explains the more concise relationship in Figure 2 (right panels), where the least resilient vegetation is found where there are both long-term water deficits (high aridity) and unreliable precipitation (high inter-annual variability). Changes in drought extremes will of course influence vegetation resilience, though it is not clear that changes in recent decades are the root cause of the scaling seen in Figure 2. Further research around local- or regional-scale droughts – rather than the globally aggregated statistics we present here – would be required to better constrain this relationship.

We have added an additional line to our “Intra- and Inter-annual precipitation variability” section on the topic of changing drought dynamics.

Below are some new specific comments:

1. This study focuses on the variability of vegetation resilience, so the vegetation data must be detrended before the analyses. The authors have mentioned it, but how they detrended their data is unclear. Some sentences can be added in the method section to describe it.

We had not included an extensive description of our de-trending and de-seasoning procedure previously, but rather a short line mentioning that we used STL and standard processing as seen in previous papers. We have now added a description of the STL method that we used in the methods section of the present manuscript, and have added a code repository in the code availability statement of the manuscript to make this clearer. This code repository also contains data needed to interpret the statistical relationships shown on each plot, as well as further steps used in the compilation of global resilience estimates. This data is available to you as a Reviewer as a Supplemental Data File, and will be published on Zenodo if the paper is accepted.

2. The illustration of Figs. 3-5 is not very clear. One reason could be the overlaps between bars. The figure legends can be shown with less information. Also, the bars were not defined in the figure captions.

We have updated these figures after your and the other reviewer’s comments. In particular, we have cut down the number of panels and moved the display of KT-values, including information on their statistical significance, to a separate panel which serves as an overall summary of all the inferred results. We now visualize both the KT of the median line and the distribution of 1000 KT values based on random choices from within each bin as a robustness check. Original plots are maintained in the Supplement for completeness, with statistics (KT values, p-values) additionally provided in CSV files for ease of use (e.g., in addition to those shown on the plot panels in the Supplement, and those now visualized as their own panel in the main manuscript Figures 3-5). This should make our findings more easily reproducible. We have also provided short code to turn any statistic (e.g., Aridity vs Resilience for Tropical Forests) into a data table. An example is included below:

Evergreen (needle)					
Environmental Metric	Resilience Metric	Kendall-Tau	KT p -value	Frac. same	KT Sign
VOD					
Aridity (PET/P)	Empirical λ	-1	0.0833	0.521	
	λ (via AC1)	0.397	0.0273	0.747	
	λ (via Var.)	0.451	0.00854	0.792	
MODIS NDVI					
Aridity (PET/P)	Empirical λ	0.388	0.000599	0.731	
	λ (via AC1)	-0.375	0.000464	0.612	
	λ (via Var.)	-0.34	0.0015	0.615	
AVHRR NDVI					
Aridity (PET/P)	Empirical λ	-1	0.0833	0.446	
	λ (via AC1)	0.471	0.000966	0.853	
	λ (via Var.)	0.471	0.000966	0.854	

Table 1: Kendall-Tau statistics for resilience vs Aridity (PET/P) in Evergreen (needle), across three data sets.

We hope that this format will make it easy to assess any of the relationships we present – we would need 110 of the above Tables to fully cover all panels we show in our MS, which seems like an unreasonable number to include in a Supplement. All data for these tables, as well as the required code, is available as a data file for the Reviewers, and will be publicly hosted on acceptance of the paper.

3. I think the method in this study could be useful for the community, but I didn't find a statement of the code availability in the current version.

Thank you, we have added a code availability statement to our revised manuscript, alongside a code and data repository holding our pre-processing code and the data needed to generate any desired tables for the statistics shown on our plots.

Reviewer #4

Summary:

This is a review of the manuscript Global Vegetation Resilience Linked to Water Availability and Variability (NCOMMS-21-44074B-Z) submitted to Nature Communications. The topic of this manuscript is well suited to the Nature Communications as it is an investigation of the climatic drivers of the resilience of ecosystems, in particular the rate of return to 'normal' after a large disturbance. This review is a review of an already revised paper and particular care was taken in considering the previous Reviewer 3's comments.

In the paper, the authors calculate resilience in three ways, as a coefficient that best fits the recovery from a large disturbance in timeseries of NDVI (2 products) and VOD. Metrics of water availability (primarily aridity) and precipitation variability (seasonal and interannual) are then used to explain global variation of vegetation resilience for all vegetation and binned by land cover type. The paper concludes that aridity is a large driver of resilience, while precipitation variability is an important but weaker driver. The paper is an interesting study of the connection of climate to resilience and the results discussed appear to be plausible. However, overall, I found the statistical treatment of the relationships between climate and resilience to be lacking or difficult to interpret. This is partly due to the difficult to read inclusion of the statistical results in the legends of figures. In addition, the concern that a change in statistics due to timeseries being made of multiple instruments could be more strongly addressed. Detailed comments below.

Thank you for taking the time to review our paper. We will endeavor to answer your specific concerns below.

Reviews of [figures] and [lines] below.

Major revisions:

[397 – 401] While the additional MODIS NDVI time series does support the robustness of the conclusions to a change in dataset, a more complete treatment of this potential issue is needed or the relevant results of the linked to study need to be detailed in the current paper and response. The link to the pre-review paper is not sufficient. If the resilience calculation were detecting either the change in instrument, or were finding significantly different resilience in one instrument time series vs. another it should appear in the time series of when the resilience events occur. One possibility is to show a distribution of when the resilience events being fit occur.

Thank you for your comment. As the results of our synthetic experiments, which investigate the effects of changing satellite sensor mixes over time on temporally resolved resilience estimators, are still in review, we did not want to delve too deeply into them in this paper. However, in response to this comment, we have added additional discussion of multi-instrument data to the MS. If our other work is published before this one, we will make it clear how the results of the two papers complement each other. The new lines in the introduction read:

Combining data from different satellite sensors can lead to biases in estimates of the temporal changes of resilience indicators due to induced time-varying changes in the higher-order statistics of the resulting times series (Smith et al., ESD). Here, however, we only compute resilience indicators over the full available time spans of each data set; in this case, a changing satellite composition does not induce systematic biases in our resilience estimates (cf. Methods).

We have also added additional paragraphs in the Methods to further explain our argument:

For the empirical estimate of the restoring rate obtained from fitting an exponential to the recovery after an abrupt negative deviation of VOD or NDVI, abrupt changes in the mean state induced by changing sensors rather than an actual vegetation shift may impact the results. However, all data sets used here are tightly cross-calibrated to eliminate mean-shifts when new instruments are introduced. It is therefore unlikely that changes in the instrumentation of the various data sets unduly influence our empirical estimates of λ .

and:

It is important to note that combining data from different sensors with varying signal-to-noise ratios (e.g., VOD, AVHRR NDVI) can bias estimates of temporal changes in resilience indicators because the higher-order statistics of the resulting time series are not homogeneous (Smith et al., 2022; ESD). In the present work, however, we do not investigate temporal trends via estimating resilience indicators in sliding windows (as in (Smith et al., 2022; ESD)), but rather estimate resilience indicators for the full available time series. This excludes the possibility of systematic biases in our AC1- and variance-based estimates of the restoring rate λ . In principle, the combination of different sensors might lead to larger uncertainties in the different methods we use to estimate λ : combining different sensor data leads to temporally varying (yet spatially homogeneous) effects on the AC1 and variance and may therefore lead to a wider spread -- but not a bias -- in the resulting estimates of λ .

To see visually if the switching of sensors leads to noticeable changes in the number of transitions, we have included the proposed histogram of 'event timing' as Figure 1 of this Reply:

Figure 1: Timing of empirical recovery events for three different RSQ cutoffs. Black bars denote instrument changes.

While we note some annual and inter-annual cyclicity in the number of transitions, they do not match with changes in instruments for either AVHRR nor VOD.

Additionally, it would be useful to know how many points or events are happening in a particular location and how many events are being screened out based on the R2 metric [397 – 401].

Most grid cells have only a single ‘transition event’ that we use for the empirical estimate of lambda, though some have up to three or four (see Figure 2 of this response letter). This number is relatively low because we are very conservative in deciding which events to consider (e.g., only events above the 99th percentile in instantaneous slope) in a given time series, see Code Repository attached as a Supplement here, and to be published on Zenodo if the paper is accepted. To make sure that the estimates are reliable, we employ several pre-selection tests, including KS-tests, checking how far the raw time series drops, and – most importantly – the fit statistic in terms of the R2 metric. We have included the number

of points for various R2 cutoffs in the above Figure 1, as well as in the statistical Tables included as Supplementary information. A map of transition locations is included as Figure 2 below:

Figure 2 – Locations of empirical recovery fits, with $R^2 > 0.2$ (i.e., what is shown in the line plots in the Manuscript).

[336 – 338][461-463][Figures 3 – 5, Supplemental Figures S4 – S11] This is a comment broadly on how statistics are used and displayed to support statements around ‘robust trends’ (here I am assuming this is the relationship between predictors and resilience, rather than through time) and ‘break points’. The display items that support the relationship between climate variables and resilience contain so much information it is very difficult to interpret. While there do appear to be relationships there, the display figures, and paper as a whole, do not concisely show the statistical uncertainty of those relationships. In addition, the statistical results contained in the legend are nearly unreadable and very difficult to cross reference. I suggest summarizing these statistics in a few tables and potentially focusing on one dataset or one resilience metric in the main text that best supports the discussion while moving the rest to the supplement.

Thank you for this comment. We endeavored to show multiple overlapping results (e.g., three resilience metrics, three sets of instruments, several land cover types) to demonstrate the robustness of our conclusions, as well as to illustrate potential differences between the data sets and metrics. However, based on your comment and that of the other Reviewer, we have updated our graphics to be more concise. In particular, we chose two data sets and two metrics (rather than 3 datasets and 3 metrics) for the new main text figures, namely the empirical and AC1-based λ estimates, for VOD and MODIS NDVI, and shifted the remaining panels to the SI. This gives us space to include information about the distribution of KT values for a given predictor-resilience relationship (e.g., Aridity vs AR1-based λ) as a box plots. We feel that the main point – the distribution of KT values for a given relationship and instrument – is concisely displayed now together with the p -value testing the significance of the KT statistic of the restoring rate λ as a function of the given predictor, in additional panels in Figures 3-5.

For completeness, we have also included all statistics (each instrument, land-cover type, resilience metric, and environmental metric, i.e., every line shown on one of our plots) as three CSV files. These files cover (1) all environmental variables, (2) the same variables but with the precipitation percentile cutoff (e.g., Supplemental Figures S11, S12), and (3) the results of changing the bin sizes for our resilience-aridity relationships (e.g., Supplemental Figures S13, S14). To make this easier to parse, we have also included python code that generates a table for any given line on one of the plots (e.g., Savanna land cover aridity vs resilience, Grasslands seasonality vs resilience, etc). We feel this is a more complete solution than including 100+ tables in the Supplement to cover all statistics presented. These tables also include additional information you requested, such as the number of data points and number of bins for each land-cover type used. All data and code can be found in the included Supplementary Data file, and will be published on Zenodo if the paper is accepted.

To better support the discussion around a break point (aridity < 1) or the relative strength of relationships between resilience and the variability of resilience it needs to be clear how much of the data is explained by a linear model (trend) or potentially a parabolic or piecewise model (break point). I found it very difficult to understand which land covers had significant relationships of resilience vs. aridity etc. and whether those relationships were statistically significant.

In order to determine the overall relationship between the different predictors (aridity, seasonality, inter-annual variability) we relied on Kendall-Tau statistics, to be able to take into account nonlinearities. As a further robustness check, we propagate the uncertainties of the estimates of λ (indicated by the vertical error bars in the upper panels of Figures 3 to 5) by randomly sampling from our bins and generating an ensemble of KT statistics. We reported how many of those random iterations of our process agree with the median (i.e., answering the question of whether all iterations match the pos/neg KT trends of the medians of each bin) on the legend of our figures. We have now included this KT distribution as box plots in the main manuscript figures instead, and added these complete statistics as data Tables.

In regards to break points, we discussed qualitative differences between regions above and below Aridity=1. Most land covers do not show such a strong difference – this is something that is visible in certain land-covers (e.g., Savanna, Evergreen Broadleaf, Grasslands, Deciduous Broadleaf); further, not all data sets show these relationships in the same way. We do not think this is a global-scale relationship

that holds true across all land covers and ecosystems, but rather something that some regions display. We have reported statistics for before/after aridity=1 for Savannas as an illustrative example, but note that their robustness is somewhat limited – two linear models over ~20 points is not a lot of data to work with. We have modified our discussion in that Section (“Long-term Water Availability”) to put less weight on this as a global-scale argument. Our main argument remains that over the range of Aridity (or precipitation variability), there is a non-linear relationship – tested with Kendall-Tau statistics – that points to decreasing resilience with less water availability and consistency.

Reviewer comments, fourth round -

Reviewer #4 (Remarks to the Author):

Comment:

This is a review of the manuscript Global Vegetation Resilience Linked to Water Availability and Variability (NCOMMS-21-44074C) submitted to Nature Communications. The authors have done an excellent job of addressing my previous comments.

As a point of clarification: In their new display charts for Figure 3, 4, 5 E, the KT value for the median curve (black triangle or dot depending on significance) is generally very different from that of the KT value calculated from random sampling. My understanding is that the boxplot (random sampling) is to provide insight into the uncertainty of the KT value, in this case I would expect the value calculated from the median to fall somewhere inside the range of the boxplot? It would help avoid confusion if there was a line in the caption as well as the paper explaining why we would or would not expect these two calculations to overlap. I broadly found these plots very successful in summarizing the data.